

# Iron fertilization efficiency and the number of past and future regenerations of iron in the ocean

Benoît Pasquier[1,2] and Mark Holzer[1]

[1]Department of Applied Mathematics, School of Mathematics and Statistics, University of New South Wales, Sydney, NSW, Australia.
[2]Now at Department of Earth System Science, University of California, Irvine, CA, USA.

**Correspondence:** Benoît Pasquier (pasquieb@uci.edu)

**Abstract.** Iron fertilization is explored by tracking dissolved iron (DFe) through its life cycle from injection by external sources (birth) to burial in the sediments (death). We develop new diagnostic equations that count iron and phosphate regenerations with each passage through the biological pump and partition the ocean's DFe concentration according to the number of its past and future regenerations. We apply these diagnostics to a family of data-constrained estimates of the iron cycle with sources $\sigma_{\text{tot}}$ in the range $1.9$–$41\,\text{Gmol yr}^{-1}$. We find that for states with $\sigma_{\text{tot}} > 7\,\text{Gmol yr}^{-1}$, $50\%$ of the DFe inventory has not been regenerated in the past and $85\%$ will not be regenerated in the future. The globally averaged mean number of past and future regenerations scale with the bulk iron lifetime $\tau \sim \sigma_{\text{tot}}^{-1}$ and have ranges of $0.05$–$2.2$ and $0.01$–$1.4$, respectively. Memory of birth location fades rapidly with each regeneration, and DFe regenerated more than $\sim 5$ times is found in a pattern shaped by Southern Ocean nutrient trapping. We quantify the natural fertilization efficiency at any point $r$ in the ocean as the global export production resulting from the DFe at $r$, per iron molecule. We show that this efficiency is closely related to the mean number of future regenerations that the iron will experience. At the surface, the natural fertilization efficiency has a global mean in the range $0.7$–$7\,\text{mol P}\,(\text{mmol Fe})^{-1}$ across our family of state estimates and is largest in the central tropical Pacific, with the Southern Ocean having comparable importance only for high iron-source scenarios.

## 1   Introduction

Iron is an essential micronutrient for phytoplankton photosynthesis (e.g., Raven, 1988). In the ocean, dissolved iron (DFe) is a trace element that has been shown to limit biological production over vast high-nutrient-low-chlorophyll (HNLC) regions (e.g., Boyd et al., 2007; Aumont and Bopp, 2006). In HNLC regions, ample phosphate and nitrate are available but phytoplankton are not able to completely utilize these macronutrients because of a lack of sufficient DFe. Iron thus exerts a major influence over the marine carbon cycle and hence over the global climate system.

Many studies have been devoted to quantifying the extent to which biological productivity and ocean carbon uptake can be influenced by altering the iron supply through geoengineering (e.g., Aumont and Bopp, 2006; Boyd et al., 2007). Relatedly, it is natural to ask how efficient iron is in supporting the export of organic matter in the current state of the system and what the relative efficiencies of the different iron sources are (e.g., aeolian versus sedimentary). Zeebe and Archer (2005) suggested that artificial iron fertilization would not sufficiently impact atmospheric $CO_2$ to mitigate anthropogenic global warming. Zeebe





and Archer (2005)'s argument relied, among other things, on quantifying the efficiency of iron fertilization, which was inferred from the Southern Ocean Iron Experiment (SOFeX). Similarly, de Baar et al. (2008) used data from localized artificial iron fertilization experiments to explore the per-iron-molecule efficiency of supporting carbon export, defined by the change in local dissolved inorganic carbon per unit added DFe. From this, de Baar et al. (2008) estimated the "natural" fertilization efficiency

(i.e., the efficiency of the unperturbed system) by correcting for the lack of ligand protection in artificial iron fertilization.

Here we develop new diagnostics for following an iron molecule's passages through the biological pump from its "birth" by source injection to its "death" by scavenging and burial in the sediments. We show that the number of times that an iron molecule is biologically utilized and regenerated at depth during its birth-to-death journey is a fundamental metric for understanding the marine biosphere's iron fertilization efficiency. Using Green-function techniques, we track DFe through past

regenerations back to its birth and through future regenerations forward to its death. We apply our new diagnostics to a family of data-constrained state estimates of the iron cycle and quantify the natural iron fertilization efficiency in three dimensions throughout the global ocean. To the best of our knowledge, this has never done before. Previous studies, like those of Zeebe and Archer (2005) and de Baar et al. (2008), estimated the fertilization efficiency only for the specific surface regions where iron was artificially injected into the euphotic zone.

Our analysis uses the 287 optimized state estimates of Pasquier and Holzer (2017), which were obtained from a steady-state inverse model of the ocean's coupled iron, phosphorus, and silicon cycles, embedded in the data-assimilated steady global ocean circulation of Primeau et al. (2013). The members of our family of state estimates correspond to different iron source strengths, given that the actual values for the real ocean are still highly uncertain (e.g., Tagliabue et al., 2016). All these optimized states fit the available observed DFe and nutrient concentrations about equally well, with a total iron source that

ranges from 1.9 to $41\,\mathrm{Gmol\,Fe\,yr}^{-1}$ across our family. (Atmospheric models estimate an aeolian source of soluble iron to the ocean of $\sim 6\,\mathrm{Gmol\,Fe\,yr}^{-1}$ (Luo et al., 2008).) Having states for a wide range of iron source scenarios enables us to quantify systematic variations with source strength.

While recent studies of the iron cycle have begun to quantify the distribution and pathways of regenerated iron (Tagliabue et al., 2014; Qin et al., 2016; Achterberg et al., 2018), to the best of our knowledge the biological cycling of iron during its full

birth-to-death lifetime has previously not been quantified. In particular, the future passages of a given DFe molecule through the biological pump have not been systematically considered, thus missing a potentially important part of its contribution to supporting carbon export. In fact, de Baar et al. (2008) dismissed the importance of iron regenerated at depth for subsequently fertilizing production altogether, based on the assumption that all DFe that is regenerated at depth would be lost to scavenging. However, scavenging is weaker at depth so that deep regenerated DFe molecules are not necessarily scavenged and instead

may be transported back to the surface where they may support production multiple times before being scavenged out of the system.

Here, for the first time, the entire birth-to-death journey of DFe is considered. Specifically, we aim to address the following questions:

1. What fraction of the DFe distribution in the current state of the ocean has passed $n$ times through the biological pump in

the past, and what fraction will pass $m$ times through the biological pump in the future, for any given $n$ and $m$?





2. How much does the DFe present at any given location in the current state of the ocean contribute to the global export production per DFe molecule?

3. How do the mean number of past and future passages through the biological pump, and the closely related iron fertilization efficiency, depend on the uncertain iron source strengths?

We find that for states with global iron sources above $7\,\mathrm{Gmol\,Fe\,yr^{-1}}$, most of the DFe gets scavenged out of the system without participating in the biological pump. Future passages through the biological pump are less likely than past passages, but for sources around $7\,\mathrm{Gmol\,Fe\,yr^{-1}}$, roughly $10\,\%$ of the iron inventory will still participate in future biological production before death. DFe that has been regenerated more than about $5$ times in the past can be found in a characteristic pattern (mathematically an eigenmode) that bears a strong signature of Southern Ocean nutrient trapping. DFe that will be regenerated many times in the future is similarly found in a Southern-Ocean-intensified eigenmode.

We link the mean number of future regenerations to the natural iron fertilization efficiency, which we are able to quantify at any point in the ocean. At the surface, which is most relevant to geoengineering, we find that this fertilization efficiency is largest in the central tropical Pacific, with the Southern Ocean having comparable efficiency only for states with a high total iron source.

In Section 2, we briefly detail the salient features of the iron model and introduce our diagnostic framework. The DFe distribution is partitioned according to the number of past and future passages through the biological pump in Sections 3 and 4, respectively. In Section 5, we develop the connection with the iron fertilization efficiency. Caveats of our approach are discussed in Section 6, and we present conclusions in Section 7.

## 2 Iron Model

### 2.1 Nonlinear Model

Following Pasquier and Holzer (2017), we write the nonlinear tracer equation for DFe concentration $\chi$ as

$$(\partial_t + \mathcal{T})\chi = \sum_c (\mathcal{S}_c - 1)U_c + \sum_j (\mathcal{S}_j - 1)J_j + \sum_k s_k \quad, \tag{1}$$

where $\mathcal{T}$ is the advective-diffusive transport operator of the data-assimilated steady ocean circulation of Primeau et al. (2013). $U_c$ is the iron uptake rate of phytoplankton functional class $c$, $J_j$ is the iron scavenging rate for particle type $j$, and $s_k$ is the source of dissolved iron of type $k$, with $k = \mathrm{A}$, S, or H, for aeolian, sedimentary, and hydrothermal iron, respectively. The subscript $c$ ranges over small, large, and diatom phytoplankton functional classes, and the subscript $j$ ranges over particulate organic phosphorus (POP), biogenic silica (BSi), and dust particle types.

The uptake $U_c$ depends nonlinearly on the concentration of DFe, phosphate, and silicic acid and hence couples the iron, phosphorus, and silicon cycles. The remineralization rate of DFe taken up by phytoplankton class $c$ is modelled by $\mathcal{S}_c U_c$, where the linear "source" operator $\mathcal{S}_c$ accomplishes both the in situ remineralization in the euphotic zone and the biogenic transport followed by remineralization at depth. (Remineralization at depth is modelled as instantaneous with the divergence



**Table 1.** Iron source strengths in units of $\mathrm{Gmol\,Fe\,yr^{-1}}$ for the typical state and four other states sampling our family of solutions. Source strengths are listed for aeolian ($\sigma_A$), sedimentary ($\sigma_S$), hydrothermal ($\sigma_H$), and total ($\sigma_{tot}$) iron. The corresponding bulk iron lifetime, $\tau$, in units of years, is also tabulated. ($\tau$ is the ratio of the global DFe inventory to $\sigma_{tot}$.)

| Source scenario | Acronym | $\sigma_A$ | $\sigma_S$ | $\sigma_H$ | $\sigma_{tot}$ | $\tau$ |
|---|---|---|---|---|---|---|
| Low-hydrothermal | LoH | 5.3 | 1.7 | 0.15 | 7.2 | 113 |
| High-sedimentary-low-aeolian | HiS-LoA | 1.8 | 6.2 | 0.87 | 8.9 | 96 |
| Typical | TYP | 5.4 | 1.7 | 0.88 | 8.0 | 104 |
| High-aeolian-low-sedimentary | HiA-LoS | 15. | 0.45 | 0.88 | 17. | 45 |
| High-hydrothermal | HiH | 6.3 | 2.0 | 2.3 | 11. | 80 |

of a Martin power-law POP flux profile (Martin et al., 1987).) Similarly, the redissolution rate of scavenged DFe is modelled by $\mathcal{S}_j J_j$ where the source operator $\mathcal{S}_j$ represents the instantaneous transport to depth of iron scavenged by particles of type $j$. The detailed model formulation of the coupled iron–phosphorus–silicon cycles, including the construction of the operators $\mathcal{S}_c$ and $\mathcal{S}_j$, has been published by Pasquier and Holzer (2017).

## 2.2 Family of Optimal State Estimates

Pasquier and Holzer (2017) coupled the iron cycle to the global phosphorus and silicon cycles and determined the steady state of the system using an efficient Newton solver. The biogeochemical model parameters were systematically optimized by minimizing the quadratic mismatch between modelled and observed nutrient and phytoplankton concentrations. This approach led to a family of 287 possible state estimates, which correspond to widely different iron-source scenarios and a range in total iron source strength of 1.9 to $41\,\mathrm{Gmol\,Fe\,yr^{-1}}$. (The true magnitude of the ocean's iron sources is still highly uncertain (e.g., Tagliabue et al., 2016).) All state estimates have very similar fidelity to the observational constraints.

By performing our analyses for a family of states spanning a wide range of possible iron sources, we are able to establish features that are insensitive to the iron-source scenario, while also being able to quantify systematic variations of key metrics with the uncertain iron source strengths. Below we will show spatial patterns for a typical state estimate and for four other states that are representative of the variations across our family of estimates. The source strengths and bulk iron lifetimes of these five representative states are collected in Table 1. When emphasizing systematic variations of a specific metric with iron source strength, we will plot the metric for all 287 states.

### 2.3 Equivalent Linear Model: Iron Labelling Tracers

In order to track iron from its birth at the source to its eventual death via the irreversible part of scavenging, we consider a tracer $\chi$ that has the same concentration as DFe, but for which the nonlinear uptake and scavenging are replaced by linear processes. These linear processes are diagnosed from the nonlinear steady-state solution of Eq. (1) to provide identical uptake and scavenging rates and hence identical tracer solutions. Specifically, we diagnose the local rate constants $\gamma_c(\boldsymbol{r}) = U_c(\boldsymbol{r})/\chi(\boldsymbol{r})$





and $\gamma_j(\boldsymbol{r}) = J_j(\boldsymbol{r})/\chi(\boldsymbol{r})$, so that $U_c$ and $J_j$ can be replaced by $\gamma_c\chi$ and $\gamma_j\chi$, which are linear in $\chi$. Following Pasquier and Holzer (2017), we write $\mathcal{S}_c = (1 - f_c) + \mathcal{B}f_c$ to separate the remineralization rate into the fraction $(1 - f_c)$ that is remineralized in situ in the euphotic zone and the detrital fraction $f_c$ that is exported to depth by the biogenic transport and remineralization operator $\mathcal{B}$. Substituting the linear forms of $U_c$ and $J_j$ into Eq. (1) and reorganizing terms, we obtain

$$(\partial_t + \mathcal{T})\chi = \mathcal{R}\chi - \mathcal{L}\chi - \mathcal{D}\chi + \sum_k s_k \quad , \tag{2}$$

where $\mathcal{L} \equiv \sum_c f_c \gamma_c$ is the uptake operator for DFe that gets exported, $\mathcal{R} \equiv \mathcal{BL}$ is the regeneration operator, and $\mathcal{D} \equiv \sum_j (1 - \mathcal{S}_j)\gamma_j$ is the reversible scavenging operator. The operator $\mathcal{D}$ represents iron scavenging minus redissolution of scavenged iron. Thus, $\mathcal{D}$ provides the death process for DFe, which is the net permanent loss due to burial in the sediments. The operator $\mathcal{B}$ represents conservative biogenic particle transport and subsequent regeneration. In Eq. (2), DFe enters the biological pump with the utilization rate $\mathcal{L}\chi$ and exits the biological pump with the regeneration rate $\mathcal{R}\chi$. Throughout, "regeneration" refers to

remineralization in the aphotic zone following a passage through the biological pump. (We define the regeneration operator $\mathcal{R}$ such that in the euphotic zone $\mathcal{R}\chi = 0$ so that the biological pump's intake ($\mathcal{L}\chi$) and output ($\mathcal{R}\chi$) are cleanly separated.)

The source $s_k$ in Eq. (2) may be thought of as injecting tracer labels attached to DFe. These labels are eventually completely removed by $\mathcal{D}$. The uptake $\mathcal{L}$ also removes labels from iron in the euphotic zone, but the regeneration $\mathcal{R} = \mathcal{BL}$ re-injects these labels throughout the water column below without any losses ($\mathcal{B}$ is conservative). A schematic of the transport and cycling of

iron labels by these operators is provided in Figure 1. For convenience, we bring all the operators to the left-hand-side of the equation so that (2) can be written compactly as $(\partial_t + \mathcal{H})\chi = \sum_k s_k$, where the complete linear system operator is given by $\mathcal{H} \equiv \mathcal{T} - \mathcal{R} + \mathcal{L} + \mathcal{D}$. Below it will be useful to consider the iron cycle without regeneration, whose evolution is governed by $\mathcal{F} \equiv \mathcal{T} + \mathcal{L} + \mathcal{D}$. (Note that $\mathcal{F} = \mathcal{H} + \mathcal{R}$.)

We emphasize that the linear equivalent model (2) has solutions that are identical to the solutions of nonlinear model (1).

The linear model (2) is not a linearization of the full nonlinear model but, instead, tracks passive labelling tracers that faithfully follow the DFe through the nonlinear biological and scavenging processes.

The linearity of the labelling tracers allows us to consider the DFe concentrations to be the sum of the concentrations of different iron "source types", that is $\chi = \sum_k \chi_k$, where $\chi_k$ is the concentration due to source $s_k$. In steady state, we can therefore calculate the concentration $\chi_k$ of each source type by solving (Holzer et al., 2016)

$$\mathcal{H}\chi_k = s_k \quad . \tag{3}$$

## 3 Past Contributions to Export

To establish how much a given iron molecule has contributed to organic-matter export since its birth, we partition the DFe concentration according to the number of its past passages through the biological pump.





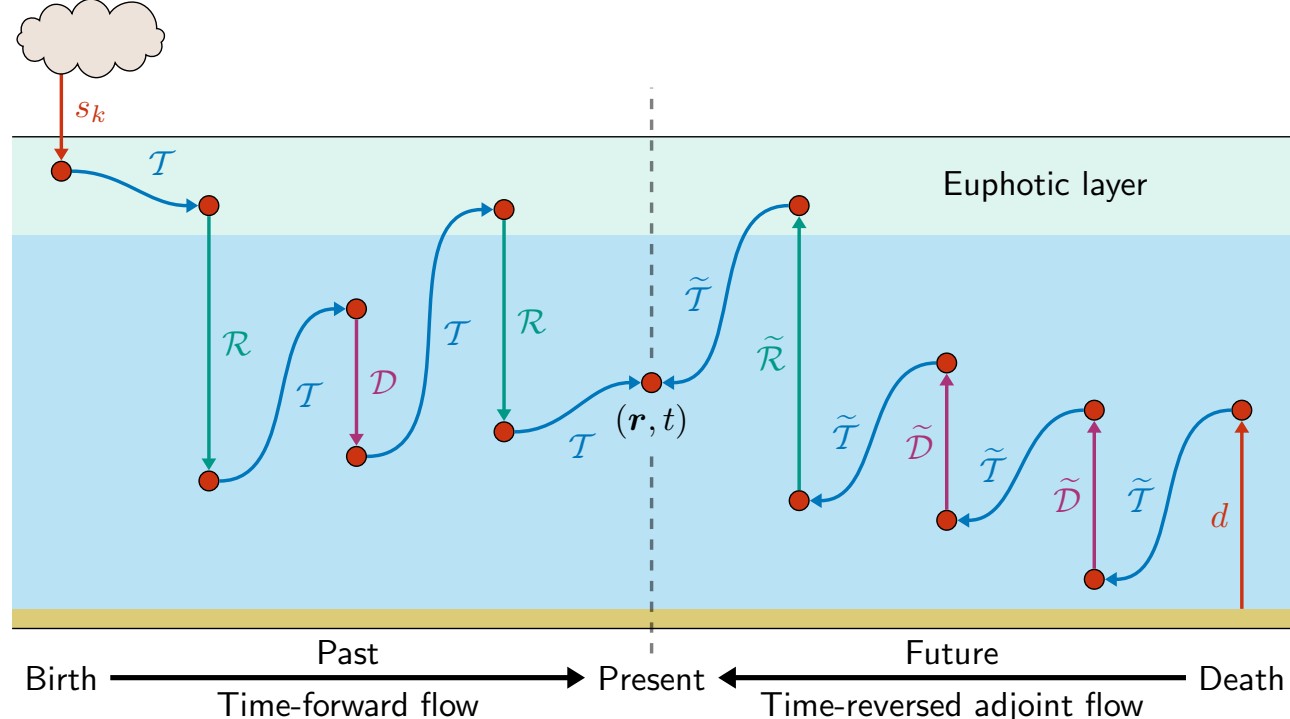

**Figure 1.** Schematic of the birth-to-death lifecycle of a labeled DFe molecule (red dot). As captured by Eq. (2), the molecule makes several passages through the biological pump (green, $\mathcal{R}$), through the scavenging pump (purple, $\mathcal{D}$) while being transported by the ocean circulation (blue, $\mathcal{T}$) from its birth (red, $s_k$, here the aeolian source) to its death by burial in sediments (red, $d$). This particular DFe molecule passed $n = 2$ times through the biological pump since its birth, and will pass $m = 1$ times through the biological pump until its death. We consider the DFe concentration at location $\boldsymbol{r}$ and at present time $t$. Our diagnostics partition DFe at $(\boldsymbol{r}, t)$ into the fractions that have undergone $n$ regenerations since birth and that will undergo $m$ regenerations until death. Computationally, we track DFe from birth to the present using the usual time-forward flow, while we track DFe backward in time from its death to the present using the adjoint operators ($\widetilde{\mathcal{T}}$, $\widetilde{\mathcal{R}}$, and so on).

### 3.1 Iron Concentration Regenerated $n$ Times Since Birth

We first consider the concentration of DFe that has never been regenerated since it was injected (born) by the source. This concentration, denoted by $\chi_k^{0\downarrow}$, can simply be computed from our equivalent linear system by injecting iron labels with source $s_k$, but not permitting them to be regenerated. We therefore remove the $\mathcal{R}$ term from Eq. (3), which is equivalent to replacing $\mathcal{H}$ with $\mathcal{F}$, to obtain

$$\mathcal{F}\chi_k^{0\downarrow} = s_k \quad . \tag{4}$$

We use arrow superscripts to indicate past processes that occurred since injection *into* ($\downarrow$) the ocean, or future processes that will occur until removal *out of* ($\uparrow$) the ocean.



We can now calculate the concentration of DFe from source $k$ that has been regenerated exactly one time, $\chi_k^{1\downarrow}$, as follows. The source of labels for $\chi_k^{1\downarrow}$ is the rate of first regeneration of DFe, which is given by $\mathcal{R}\chi_k^{0\downarrow}$. We simply allow the system to cycle these labels but remove them on uptake using $\mathcal{F}$ (no second regeneration). Thus, in steady state, $\chi_k^{1\downarrow}$ obeys $\mathcal{F}\chi_k^{1\downarrow} = \mathcal{R}\chi_k^{0\downarrow}$. Similarly, the source of DFe that has been regenerated exactly $n+1$ times since birth is the rate of $(n+1)$st regeneration. This gives the recursion relation

$$\mathcal{F}\chi_k^{(n+1)\downarrow} = \mathcal{R}\chi_k^{n\downarrow} \quad , \tag{5}$$

with (4) providing the starting point of the recursion. Note that the $\chi_k^{n\downarrow}$ partition the DFe concentration exactly, with $\sum_{n=0}^{\infty} \chi_k^{n\downarrow} = \chi_k$, as shown in Appendix B.

Figure 2 shows global zonal averages of the iron concentration for our typical state, partitioned by source type (aeolian, sedimentary, hydrothermal) and according to the number $n$ of past regenerations since birth. To emphasize the spatial patterns, each $\chi_k^{n\downarrow}(\boldsymbol{r})$ field has been normalized by its global volume-weighted average, $\langle\chi_k^{n\downarrow}\rangle$. Iron that has never been regenerated ($n = 0$) carries a strong source signature, with peak concentrations where the largest sources are. The patterns change dramatically with one passage through the biological pump ($n = 1$). Because iron is regenerated at depth with a Martin power-law profile, the peak zonally averaged aeolian DFe concentration that has been regenerated $n = 1$ times has a subsurface maximum at roughly $1500\,\mathrm{m}$ depth. Similarly, sedimentary iron that passed once through the biological pump has a mid-depth maximum, and is no longer concentrated near the sea floor. Sedimentary iron that has been regenerated $n = 1$ times is concentrated into the Arctic, presumably because the model has already shallow sedimentary sources there. However, the model's Arctic circulation is poorly constrained (Primeau et al., 2013) and this particular feature may not be robust. Hydrothermal iron that has been regenerated $n = 1$ times shows the signature of Southern Ocean nutrient trapping (Primeau et al., 2013; Holzer et al., 2014) as expected from the fact that hydrothermal iron is injected at seawater densities that outcrop in the Southern Ocean.

As the number $n$ of past regenerations increases, $\chi_k^{n\downarrow}$ rapidly converges to a pattern that is independent of source type. Physically, this is because the memory of birth place quickly fades with each passage through the biological pump. Mathematically, this can be seen from recursion relation (5), which gives $\chi_k^{n\downarrow} = \mathcal{A}^n \chi_k^{0\downarrow}$, where $\mathcal{A} \equiv \mathcal{F}^{-1}\mathcal{R}$. Thus, for sufficiently large $n$, the concentration $\chi_k^{n\downarrow}$ becomes proportional to the eigenmode of $\mathcal{A}$ with the largest eigenvalue $\lambda$, and the amplitude of the pattern decays exponentially like $\lambda^n = \exp(-n/n^*)$, where $n^* \equiv -1/\log(\lambda)$. (Note that $\lambda < 1$, as must be the case for $\sum_{n=0}^{\infty} \chi_k^{n\downarrow}$ to converge to $\chi_k$.) In other words, as $\mathcal{A}$ is applied repeatedly, only the projection of $\chi_k^{0\downarrow}$ on the eigenmode of $\mathcal{A}$ with the gravest eigenvalue survives. Because $\mathcal{A}$ is independent of source type, all source types approach the same large-$n$ asymptotic eigen pattern. Convergence to this pattern is remarkably rapid: For aeolian and sedimentary iron, the pattern has emerged after about 5 regenerations (not shown), and for hydrothermal iron even sooner. By $n = 10$ the asymptotic pattern is well established and indistinguishable for the different iron source types (bottom plots of Figure 2).

How robust is the large-$n$ eigen pattern of $\chi_k^{n\downarrow}$ across our family of solutions? To quantify the approach to the exponentially decaying eigenmode, we plot in Figure 3a the global mean concentration that has been regenerated $n$ times since birth as a function of $n$ for the 5 representative state estimates of Table 1. Figure 3a shows that the $e$-folding scale $n^*$ is independent of source type as expected (same large-$n$ slope for all source types on the semilog plots). For all 5 states the convergence



Figure 2. Global zonal averages of the concentration of DFe that was regenerated $n$ times since birth, normalized by its global mean, for $n = 0, 1, 2, 3$, and $10$. The normalized concentrations of the three source types (aeolian, sedimentary, hydrothermal) are shown as well as their sum, the total concentration regardless of source type. This figure is for our typical state.

to exponential decay is quickest for hydrothermal iron. The value of the $e$-folding scale $n^*$ depends on the state of the iron cycle. For our 5 representative states, the smallest $n^*$ values occur for the high-source states ($n^* \sim 0.6$ for HiA-LoS and $0.8$ for HiH), while the largest $e$-folding scales occur for the low-source states ($1.4$ for both HiS-LoA and LoH), with $n^* \sim 1.2$ for our typical state. Across the entire family of state estimates, $n^*$ ranges from about $0.4$ to $2.8$. The dependence of $n^*$ on





source scenario makes sense when we consider that all these states are optimized against the observed DFe concentrations: High aeolian input must be countered by vigorous removal from the surface ocean, and hence a large proportion of the iron in the ocean is accounted for by the first few regenerations since birth, leaving little DFe for multiple regenerations. Conversely, low aeolian input corresponds to less vigorous biological cycling so that a larger portion of the iron is available to pass through

the biological pump repeatedly. Similar consideration hold for hydrothermal and sedimentary iron. For all states, asymptotic behavior is well established for $n \gtrsim 5$.

The sedimentary and hydrothermal iron concentrations get their largest contributions from iron that has never been regenerated ($n = 0$), again pointing to significant permanent removal of these iron types before they can reach the euphotic zone following injection (birth) at depth. For aeolian iron in the typical, LoH, and HiS-LoA states, the largest contribution comes

from iron that has been regenerated $n = 1$ times as all freshly born aeolian iron is immediately available for uptake and regeneration. However, for the high-source states (HiA-LoS and HiH), the largest contribution still comes from iron that was never regenerated because the increased scavenging necessary to balance the high sources prevents most aeolian DFe from being utilized even though it is deposited directly into the euphotic zone.

Figure 3a shows that DFe that has not passed through the biological pump since birth ($n = 0$, unused iron) generally has the

highest global mean concentration, except for aeolian DFe under some source scenarios. To quantify the amount of iron that was not regenerated in the past, we now ask how the unused fraction $\int \mathrm{d}^3 \boldsymbol{r} \chi_k^{0\downarrow}(\boldsymbol{r}) / \int \mathrm{d}^3 \boldsymbol{r} \chi_k(\boldsymbol{r})$ of the global DFe inventory varies with total iron source strength, $\sigma_{\mathrm{tot}}$. This fraction may be written as $\langle f_k^{0\downarrow} \rangle_{\chi_k} = \int \mathrm{d}^3 \boldsymbol{r} f_k^{0\downarrow}(\boldsymbol{r}) \chi_k(\boldsymbol{r}) / \int \mathrm{d}^3 \boldsymbol{r} \chi_k(\boldsymbol{r})$ and therefore can be considered to be the $\chi_k$-weighted global average of the local unused fraction $f_k^{0\downarrow}(\boldsymbol{r}) \equiv \chi_k^{0\downarrow}(\boldsymbol{r}) / \chi_k(\boldsymbol{r})$. The unused fractional DFe inventory regardless of source type is given by $\langle f^{0\downarrow} \rangle_\chi$, where $f^{0\downarrow}(\boldsymbol{r}) \equiv \sum_k \chi_k^{0\downarrow}(\boldsymbol{r}) / \chi(\boldsymbol{r})$.

Figure 4a shows the fractional unused DFe inventories $\langle f_k^{0\downarrow} \rangle_{\chi_k}$ and $\langle f^{0\downarrow} \rangle_\chi$ as a function of $\sigma_{\mathrm{tot}}$ for every member of our family of state estimates. We emphasize that Figure 4 does *not* show the response of the iron cycle to changes in $\sigma_{\mathrm{tot}}$, but instead shows the fraction of the unused DFe inventory for distinct equilibrium states, each of which was optimized against observations under a different prescribed iron source scenario (Pasquier and Holzer, 2017). Because the total iron inventory is well constrained across the family, the bulk iron lifetime $\tau$, given by the ratio of inventory to source (see also Table 1), is

inversely proportional to $\sigma_{\mathrm{tot}}$, and hence the mean iron age also scales with $(\sigma_{\mathrm{tot}})^{-1}$ (Holzer et al., 2016). The systematic increase of the fractional inventory of unused iron, and its approach to saturation at $100\,\%$ for the largest sources seen in Figure 4a, reflects the fact that the probability of past regeneration decreases with the mean iron age until the age becomes so short that very little iron has a chance to pass through the biological pump before having to be scavenged out of the system to match the observed DFe concentrations and inventory. For our smallest total source of $\sigma_{\mathrm{tot}} \sim 2\,\mathrm{Gmol\,Fe\,yr}^{-1}$, DFe

molecules have lived long enough to be regenerated at least once with $\langle f^{0\downarrow} \rangle_\chi \sim 35\,\%$, while for our largest total source of $\sigma_{\mathrm{tot}} \sim 40\,\mathrm{Gmol\,Fe\,yr}^{-1}$, most DFe molecules have not had sufficient time to be regenerated and $\langle f^{0\downarrow} \rangle_\chi \sim 95\,\%$. For state estimates with $\sigma_{\mathrm{tot}} > 7\,\mathrm{Gmol\,Fe\,yr}^{-1}$, more than half of the DFe in the ocean has never passed through the biological pump, i.e., $\langle f^{0\downarrow} \rangle_\chi > 50\,\%$.

Figure 4a also shows that the fraction $\langle f_k^{0\downarrow} \rangle_{\chi_k}$ varies significantly with iron source type $k$. This is because the probability of

a past passage through the biological pump is strongly dependent on birth location. The fraction that has not been regenerated





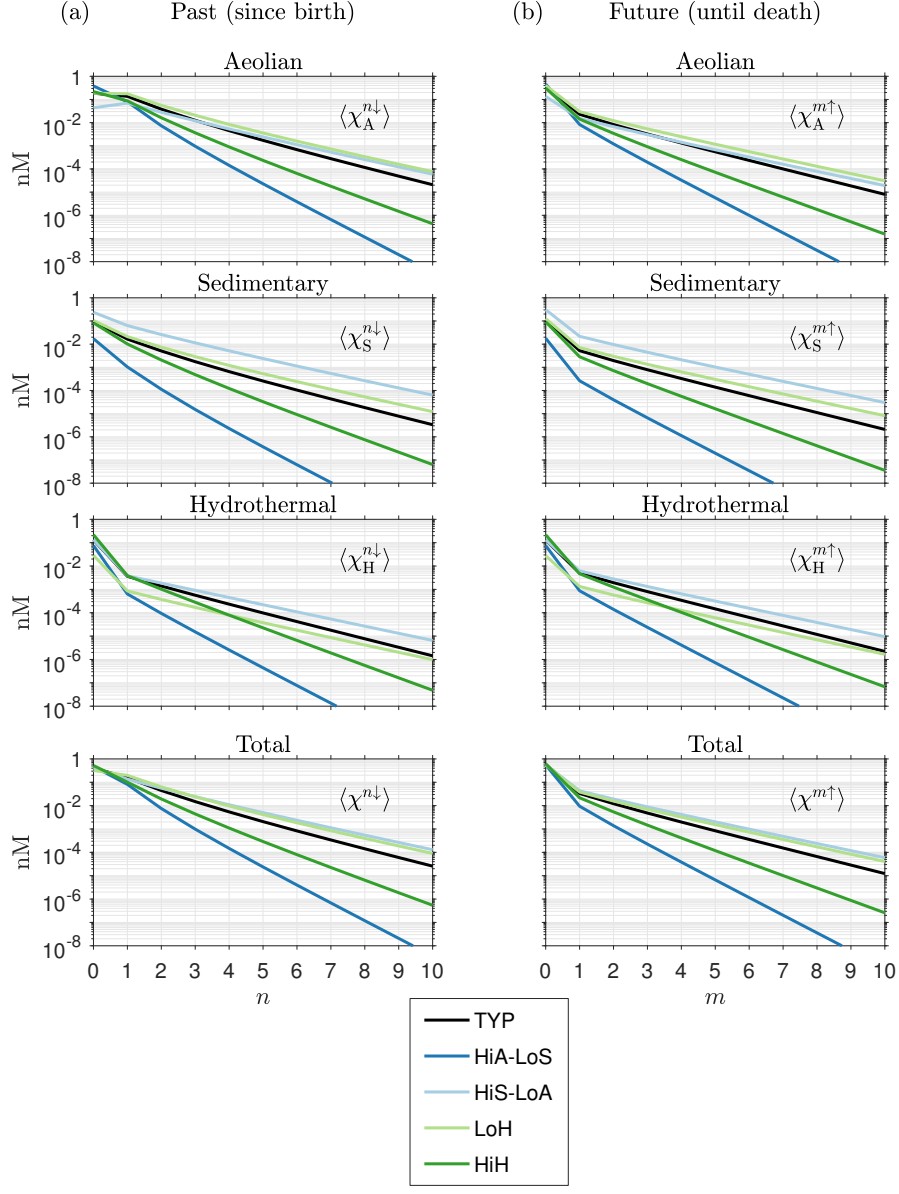

**Figure 3.** (a) Global mean DFe concentration regenerated $n$ times in the past, $\langle \chi_k^{n\downarrow} \rangle$, as a function of $n$, for five representative states of the iron cycle. (b) Mean concentration regenerated $m$ times in the future, $\langle \chi_k^{m\uparrow} \rangle$, as a function of $m$.

since birth is lower for aeolian DFe than for benthic DFe. Benthic DFe must first be transported to the euphotic zone to partici-

pate in biological production and is therefore more likely to be scavenged en route compared to aeolian DFe, which is directly

injected into the surface. Hydrothermal DFe is the least likely to have passed through the biological pump, with $\langle f_H^{0\downarrow} \rangle_{\chi_H} \gtrsim 80\,\%$

for all states and $\langle f_H^{0\downarrow} \rangle_{\chi_H} \gtrsim 95\,\%$ for states with $\sigma_{\text{tot}} > 7\,\text{Gmol Fe yr}^{-1}$. For sedimentary DFe, $\langle f_S^{0\downarrow} \rangle_{\chi_S} \gtrsim 45\,\%$ for all states,





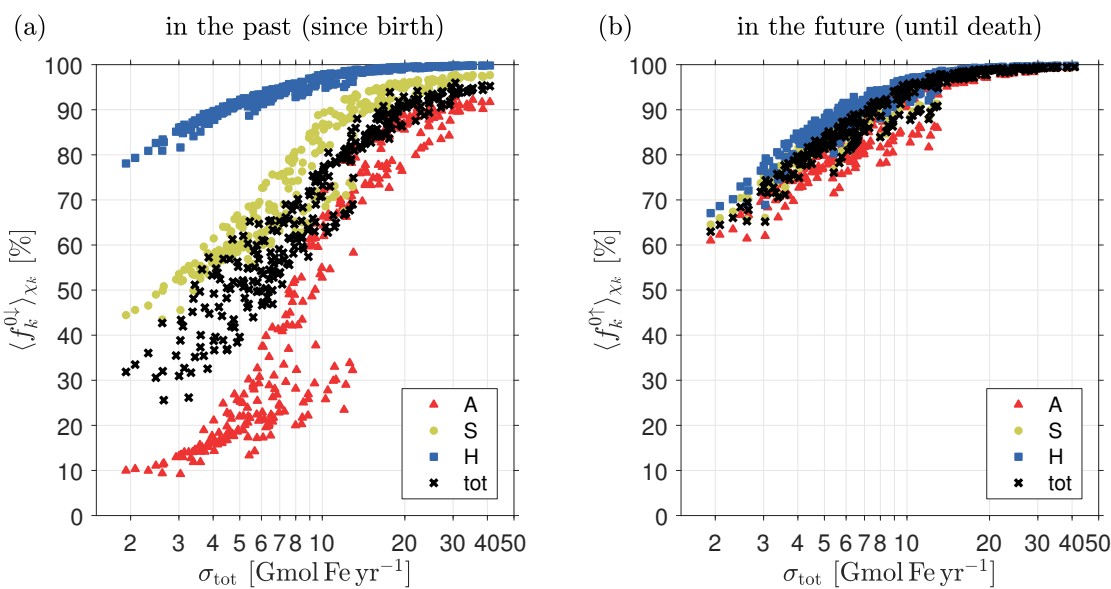

**Figure 4.** (a) Percentage of the total inventory of DFe that has not passed through the biological pump in the past (i.e., since birth), as a function of total source strength, $\sigma_{\mathrm{tot}}$, for the family of state estimates of Pasquier and Holzer (2017). (b) Percentage of the total inventory of DFe that will not pass through the biological pump in the future (i.e., until death). Note the logarithmic abscissa. Shown are the fractional inventories of the individual source types of DFe (color coded), and well as the total fractional inventory regardless of source type (black).

and $\langle f_{\mathrm{S}}^{0\downarrow}\rangle_{\chi_{\mathrm{S}}} \gtrsim 60\,\%$ for states with $\sigma_{\mathrm{tot}} > 7\,\mathrm{Gmol\,Fe\,yr^{-1}}$. As the mean age becomes ever smaller with increasing $\sigma_{\mathrm{tot}}$, the unused fractional benthic DFe inventory saturates faster to $100\,\%$ than the unused fractional aeolian DFe inventory, which reaches only $\sim 90\,\%$ even for $\sigma_{\mathrm{tot}} \sim 40\,\mathrm{Gmol\,Fe\,yr^{-1}}$, again reflecting greater probability of biological utilization for surface-injected iron. We will return to Figure 4b below where we explore future passages through the biological pump.

To explore the variations of the asymptotic eigenmode pattern across our different state estimates, Figure 5 shows the zonally averaged patterns of $\chi^{n\downarrow} = \sum_k \chi_k^{n\downarrow}$ for $n = 10$, which is well within the asymptotic regime, for our 5 representative states. (Recall that individual source types all converge to the same pattern, which is hence also the pattern of the total iron concentration.) While there is a signature of Southern Ocean nutrient trapping for all states, there is also significant variation across the state estimates. Broadly, the Southern Ocean trapping has a stronger influence on the eigenmodes of $\mathcal{F}^{-1}\mathcal{R}$ for the

high-source states than for the low-source states. This is likely due to the fact that high-source states are able to match the observed DFe concentrations by having both more active biological iron pumps and scavenging. The transport to depth by both the biological pump and by scavenging particles (the "scavenging pump") enhances the Southern Ocean trapping. The scavenging is largest at tropical and high latitudes, so that stronger scavenging will tend to remove iron from low and high latitudes (see Appendix A for zonally averaged death rates). At high latitudes this is counteracted by stronger trapping, but

at low latitudes the death rate dominates and the concentration of iron that passes many times through the biological pump





is strongly diminished. Conversely, for the low-source states, reduced scavenging allows iron to pass many times through the biological pump without being scavenged out of the system at low latitudes. Consequently, the low-source states have iron that has been regenerated many times flowing out of the Southern Ocean with mode and intermediate waters well into the Northern Hemisphere.

### 3.2   Mean Number of Regenerations Since Birth

A key metric of how much the DFe field has contributed in the past to organic-matter export is the mean number of times, $\overline{n}_k$, that a given iron molecule has been regenerated since its birth. From the local fraction of the total DFe that was regenerated $n$ times since birth, $f_k^{n\downarrow}(\boldsymbol{r})$, the mean number of past regenerations is by definition given by $\overline{n}_k(\boldsymbol{r}) \equiv \sum_{n=0}^{\infty} n f_k^{n\downarrow}(\boldsymbol{r})$. As shown in Appendix B, it follows from Eq. (5) that $\overline{n}_k$ obeys

$$\mathcal{H}(\overline{n}_k \chi_k) = \mathcal{R}\chi_k \quad . \tag{6}$$

Equation (6) may be interpreted as the equation for a labelling tracer $\overline{n}_k \chi_k$ that is cycled just like DFe by the $\mathcal{H}$ operator, but whose numerical value accumulates $\overline{n}_k$-fold with $\overline{n}_k$ past regenerations. Solving (6) provides an efficient means of finding $\overline{n}_k$ that avoids first finding and explicitly summing $f_k^{n\downarrow}(\boldsymbol{r})$.

A given iron molecule at $\boldsymbol{r}$ is responsible, on average, for the export of $\overline{n}_k(\boldsymbol{r})$ iron molecules of source type $k$ since its birth. The corresponding organic-matter export is quantified by the mean number of phosphorus molecules that are exported along

with the iron. The recycling operator for phosphorus is given by $\mathcal{R}^{\mathrm{P}} = \sum_c \mathcal{B} f_c \gamma_c^{\mathrm{P}}$ where $\gamma_c^{\mathrm{P}}(\boldsymbol{r}) \equiv U_c^{\mathrm{P}}(\boldsymbol{r})/\chi(\boldsymbol{r})$ is diagnosed from the optimized phosphorus uptake $U_c^{\mathrm{P}}(\boldsymbol{r})$ of the underlying model (Pasquier and Holzer, 2017). The mean number of phosphorus molecules, $\overline{n}_k^{\mathrm{P}}(\boldsymbol{r})$, globally exported and remineralized in the past, per DFe molecule that is currently at $\boldsymbol{r}$, obeys

$$\mathcal{H}(\overline{n}_k^{\mathrm{P}} \chi_k) = \mathcal{R}^{\mathrm{P}} \chi_k \quad . \tag{7}$$

In other words, on average, one of the type-$k$ DFe molecule currently at $\boldsymbol{r}$ has supported the export of $\overline{n}_k^{\mathrm{P}}(\boldsymbol{r})$ phosphorus molecules since its birth. In analogy with equation (6), we interpret (7) as defining a labelling tracer $\overline{n}_k^{\mathrm{P}} \chi_k$ that is cycled like

DFe by the $\mathcal{H}$ operator, but whose numerical value counts the number of phosphate molecules that were regenerated in the past via the phosphate regeneration operator $\mathcal{R}^{\mathrm{P}}$.

Figure 6 shows the global zonal averages of $\overline{n}_k(\boldsymbol{r})$ for our typical state estimate. Note that the maximum values of $\overline{n}_k$ are order unity and, on average, most of the iron in the ocean has been regenerated less than once since its birth. Iron with the largest $\overline{n}_k$ is found near the Southern Ocean surface and in mode and intermediate waters flowing out of the Southern Ocean, which is presumably a signature of Southern Ocean nutrient trapping. In terms of source types, aeolian iron has been most

active in the biological pump with $\overline{n}_{\mathrm{A}}$ exceeding $0.6$ throughout most of the ocean interior and exceeding $1.0$ throughout much of the Southern Ocean water column. In the surface waters of the low-production subtropical gyres $\overline{n}_{\mathrm{A}}$ approaches zero. The mean number of regenerations since birth is much smaller for sedimentary and hydrothermal iron, again reflecting the greater scavenging hazard for benthic iron. For sedimentary and hydrothermal iron, values of $\overline{n}_k$ greater than $0.5$ extend from the

subantarctic Southern Ocean surface into mode and intermediate waters where scavenging is relatively weak (Appendix A).



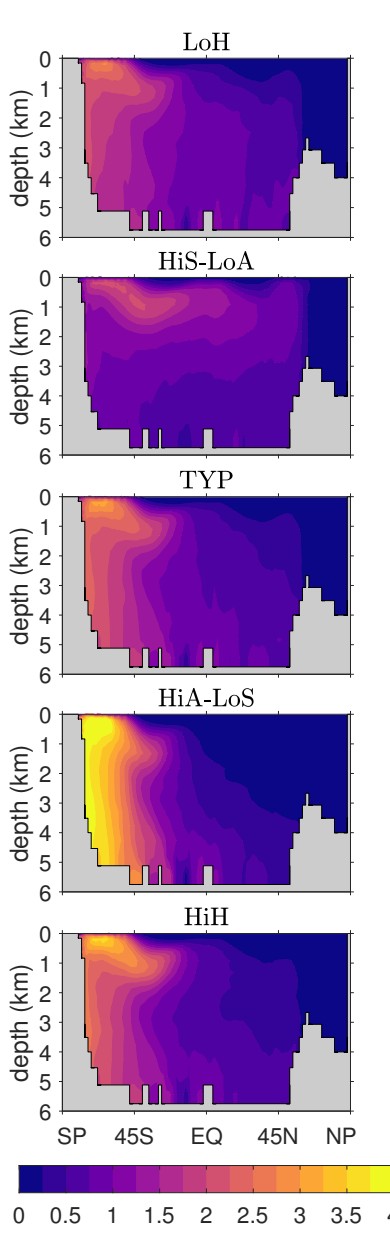

**Figure 5.** Normalized global zonal averages of the concentration of DFe that has passed $n = 10$ times through the biological pump since birth, regardless of source type, $\chi^{10\downarrow}$, to show the eigen patterns of our five representative state estimates (Table 1). The zonal averages are normalized by the global mean.



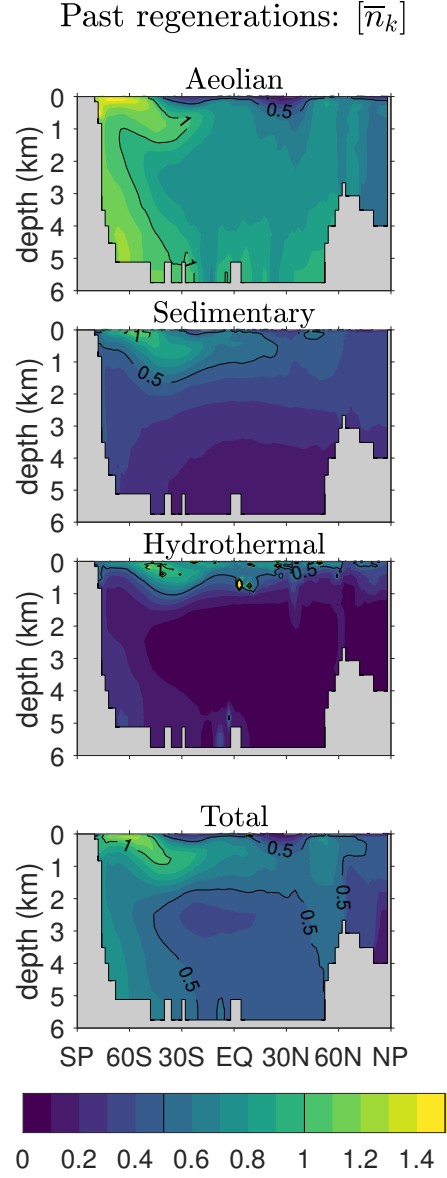

**Figure 6.** Zonal averages of the mean number of past passages through the biological pump (i.e., since birth), $\overline{n}_k$. This figure is for our typical state.

The zonally averaged *patterns* of $\overline{n}_k$ are remarkably similar across our family of estimates. As shown in Appendix C, the spatial patterns are also insensitive to the value of the recyclable fraction of scavenged iron, $f_{\mathrm{rec}}$, although their amplitude can vary by a factor of ~2 between the $f_{\mathrm{rec}} = 0$ and $f_{\mathrm{rec}} = 1$ extremes. The patterns of the global zonal averages of $\overline{n}_k^{\mathrm{P}}$ (not shown) are nearly identical to those of $\overline{n}_k$.



We expect significant variations in the amplitudes of $\overline{n}_k$ and $\overline{n}$ (the mean number regardless of source type) across the family of state estimates. Because the number of regenerations that are possible during the lifetime of an iron molecule should be proportional to the bulk iron lifetime $\tau \propto \sigma_{\mathrm{tot}}^{-1}$, we expect $\overline{n} \propto \sigma_{\mathrm{tot}}^{-1}$. To test this and to quantify the range of possible variations, Figure 7a shows the global average $\langle \overline{n} \rangle$ as a function of $\sigma_{\mathrm{tot}}$ on a log-log plot and Figure 7b shows the corresponding

behavior for $\langle \overline{n}^{\mathrm{P}} \rangle$. Both $\langle \overline{n} \rangle$ and $\langle \overline{n}^{\mathrm{P}} \rangle$ exhibit the expected approximate inverse relation with $\sigma_{\mathrm{tot}}$. Note that the $\langle \overline{n} \rangle \propto (\sigma_{\mathrm{tot}})^{-1}$ scaling is not exact; instead Figure 7a suggests that there are $(\sigma_{\mathrm{tot}})^{-1}$ scaling regimes for a low-source cluster and for a high-source cluster, with a transition for intermediate source strengths. This approximate nature of the scaling reflects the fact that the timescale for a passage through the biological pump is not merely set by the prescribed circulation and (in our model) instant particle transport and remineralization, but also depends on the spatial distribution of the scavenging, which

varies with the optimized source scenarios. The detailed timescales that link circulation, scavenging, and pumping frequency, will be explored in a future publication. The numerical values of $\langle \overline{n} \rangle$ range from 0.05 for $\sigma_{\mathrm{tot}} = 41\,\mathrm{Gmol\,yr}^{-1}$ to 2.2 for $\sigma_{\mathrm{tot}} = 1.9\,\mathrm{Gmol\,yr}^{-1}$, while $\langle \overline{n}^{\mathrm{P}} \rangle$ correspondingly ranges from approximately 0.4 to 4.3 $\mathrm{mol\,P\,(mmol\,Fe)}^{-1}$ (i.e., from 40 to 460 $\mathrm{mol\,C\,(mmol\,Fe)}^{-1}$ using a simple uniform Redfield C:P ratio of 106:1 for unit conversion).

## 4   Future Contributions to Export

We now ask how many times a given DFe molecule in the ocean will get regenerated in the future before eventually being permanently scavenged out of the system. The natural way to formulate the necessary equations is to consider the time-reversed adjoint flow (Holzer and Hall, 2000), for which the system is governed by the adjoint operators $\widetilde{\mathcal{H}}$, $\widetilde{\mathcal{F}}$, $\widetilde{\mathcal{R}}$, and $\widetilde{\mathcal{D}}$. These adjoints are defined for the volume weighted inner product. In the time-reversed adjoint flow, the death operator becomes a source of labels that we then track through sequential regenerations (see Figure 1) analogously to what we did in the previous section for

the usual time-forward flow. Here we provide the key equations — their derivation in terms of Green functions is detailed in Appendix D.

### 4.1   Number of Regenerations Until Death

To compute the number of future regenerations, it is useful to calculate the specific death rate $d(\boldsymbol{r})$, and its linear equivalent $\gamma^{\mathcal{D}}(\boldsymbol{r})\chi(\boldsymbol{r})$, with which iron is permanently (non-reversibly) removed at point $\boldsymbol{r}$. This death rate can be diagnosed from the

scavenging operator $\mathcal{D}$ and the DFe concentration $\chi(\boldsymbol{r})$ as detailed in Appendix A, where we also show basin zonal averages of $d(\boldsymbol{r})$.

One can show that the local fraction $f^{\uparrow}(\boldsymbol{r})$ that eventually dies obeys

$$\widetilde{\mathcal{H}}f^{\uparrow} = \gamma^{\mathcal{D}} \quad , \tag{8}$$

where $f^{\uparrow}(\boldsymbol{r}) = 1$ uniformly everywhere because all the DFe at any point must eventually be scavenged out of the system. In analogy with (4), the fraction $f^{0\uparrow}$ that is regenerated zero times until death obeys

$$\widetilde{\mathcal{F}}f^{0\uparrow} = \gamma^{\mathcal{D}}. \tag{9}$$

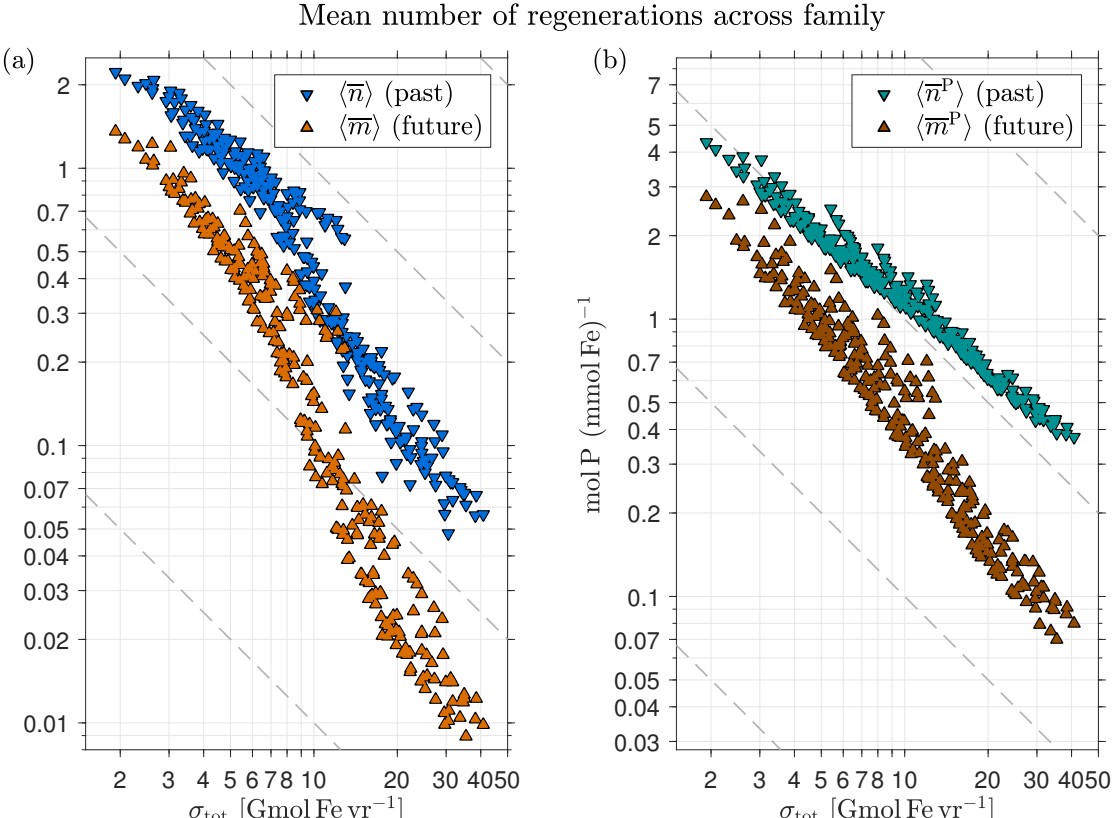

**Figure 7.** (a) The globally averaged mean number of past and future DFe passages through the biological pump per injected DFe molecule (regardless of source type), $\langle \overline{n} \rangle$ and $\langle \overline{m} \rangle$, as a function of the total iron source strength, $\sigma_{\rm tot}$, for the family of state estimates of Pasquier and Holzer (2017). (b) The corresponding globally averaged mean number of phosphorus molecules exported per DFe molecule, $\langle \overline{n}^{\rm P} \rangle$ and $\langle \overline{m}^{\rm P} \rangle$. Note the log-log axes to highlight the approximate inverse relationship with $\sigma_{\rm tot}$. To guide the eye, dashed grey lines indicate an exact $(\sigma_{\rm tot})^{-1}$ power law.

By using the regeneration rate of DFe that will be regenerated $m$ times until death and going back in time by one regeneration, we obtain the fraction that will be regenerated $(m+1)$ times until death, giving us the recursion relation

$$\widetilde{\mathcal{F}} f^{(m+1)\uparrow} = \widetilde{\mathcal{R}} f^{m\uparrow} \quad . \tag{10}$$

Note that $\sum_{m=0}^{\infty} f^{m\uparrow}(\boldsymbol{r}) = f^{\uparrow} = 1$ as can be verified from (9) and (10). The mean number of regenerations until death, $\overline{m} = \sum_{m=1}^{\infty} m f^{m\uparrow}$, can be shown to obey

$$\widetilde{\mathcal{H}} \overline{m} = \widetilde{\mathcal{R}} f^{\uparrow} \quad . \tag{11}$$



The mean number of phosphorus molecules, $\overline{m}^{\mathrm{P}}(\boldsymbol{r})$, globally exported and remineralized in the future, per DFe molecule that is currently at $\boldsymbol{r}$, obeys

$$\widetilde{\mathcal{H}}(\overline{m}^{\mathrm{P}}) = \widetilde{\mathcal{R}}^{\mathrm{P}} f^{\uparrow} \quad . \tag{12}$$

In other words, on average, one of the DFe molecule currently at $\boldsymbol{r}$, will support the export of $\overline{m}^{\mathrm{P}}$ phosphorus molecules until it is buried in the sediments.

Figure 8 shows the zonally averaged fraction $f^{m\uparrow}$ of DFe that will undergo $m$ regenerations in the future, normalized by the global average, $\langle f^{m\uparrow}\rangle$, to emphasize changes in the pattern with $m$. Note that this fraction is the same for all source types because the future of a given DFe molecule is independent of its past. Below the thermocline, the pattern of $f^{0\uparrow}$ is nearly uniform at a value just above its global mean but drops to below $50\,\%$ of the global mean near the surface where the probability of further biological utilizations and regeneration is largest (careful inspection is required to see this in Figure 8).

The pattern of $f^{1\uparrow}$ has its largest amplitude at the surface and in the Southern Hemisphere, where it spreads deepest from the surface, reflecting the relatively low Southern-Hemisphere death rates (Appendix A). As $m$ increases, $f^{m\uparrow}$ becomes rapidly proportional to the gravest eigenmode of $\widetilde{\mathcal{F}}^{-1}\widetilde{\mathcal{R}}$ [cf., Equation (10)]. As this eigenmode is approached, the pattern of $f^{m\uparrow}$ contracts into the Southern Ocean, presumably because iron that will be regenerated many times before death can only be found where there is both a low death rate and efficient nutrient trapping.

To quantify the approach of $f^{m\uparrow}$ to the gravest eigenmode of $\widetilde{\mathcal{F}}^{-1}\widetilde{\mathcal{R}}$, we return to Figure 3b, which shows the global averages of the corresponding DFe concentrations $\chi_k^{m\uparrow} \equiv \chi_k f^{m\uparrow}$ as a function of $m$. Note that $\widetilde{\mathcal{F}}^{-1}\widetilde{\mathcal{R}}$ and $\mathcal{F}^{-1}\mathcal{R}$ have the same eigenvalues so that $\langle\chi_k^{m\uparrow}\rangle$ and $\langle\chi_k^{n\downarrow}\rangle$ approach the same exponential decay (Figures 3a and 3b).

We return to Figure 4b to consider the fractional DFe inventories that will not pass through the biological pump in the future for all our state estimates. These inventories for source type $k$ and regardless of source type are given by $\langle f^{0\uparrow}\rangle_{\chi_k}$ and

$\langle f^{0\uparrow}\rangle_{\chi}$. We expect the probability of future regenerations to increase with the remaining lifetime of DFe, which should also scale like $\sigma_{\mathrm{tot}}^{-1}$ given a well-constrained global DFe inventory. This is confirmed in Figure 4b by the systematic increase and approach to $100\,\%$ saturation of $\langle f^{0\uparrow}\rangle_{\chi_k}$ and $\langle f^{0\uparrow}\rangle_{\chi}$ with increasing $\sigma_{\mathrm{tot}}$. (Note that in the theoretical $\sigma_{\mathrm{tot}} \to 0$ limit, we expect $\langle f^{0\uparrow}\rangle_{\chi_k} \to 0$ and $\langle f^{0\uparrow}\rangle_{\chi} \to 0$, although our lowest sources are not nearly small enough to exhibit this limiting behavior.)

Figure 4 shows striking asymmetries between unused past and future inventories. The fractional inventory of iron that will

not be utilized in the future (Figure 4b) is nearly independent of iron type $k$, in sharp contrast with the fractional inventory that was not utilized in the past (Figure 4a). The insensitivity of $\langle f^{0\uparrow}\rangle_{\chi_k}$ to source type is due to the independence of $f^{0\uparrow}$ on source type so that the $\chi_k$-weighted global average is only sensitive to changes in the *pattern* of $\chi_k$, which varies little across the family of states (Pasquier and Holzer, 2017). Note that for a given $\sigma_{\mathrm{tot}}$, the inventory of total DFe unused in the future, $\langle f^{0\uparrow}\rangle_{\chi}$, is significantly larger than the inventory of total DFe unused in the past, $\langle f^{0\downarrow}\rangle_{\chi}$. In other words, total DFe is more

likely to have been regenerated in the past than it is to be regenerated in the future. This asymmetry stems from the fact that $f^{0\downarrow} = \sum_k \frac{\chi_k}{\chi} f_k^{0\downarrow}$ is dominated by the relatively small unused aeolian fraction $f_{\mathrm{A}}^{0\downarrow}$ (cf. Figure 4a). In terms of individual source types, aeolian and sedimentary DFe are also more likely to have been regenerated in the past than in the future, but the reverse is true for hydrothermal DFe for which the scavenging hazard following birth is greatest. Thus, for a hypothetical state in which





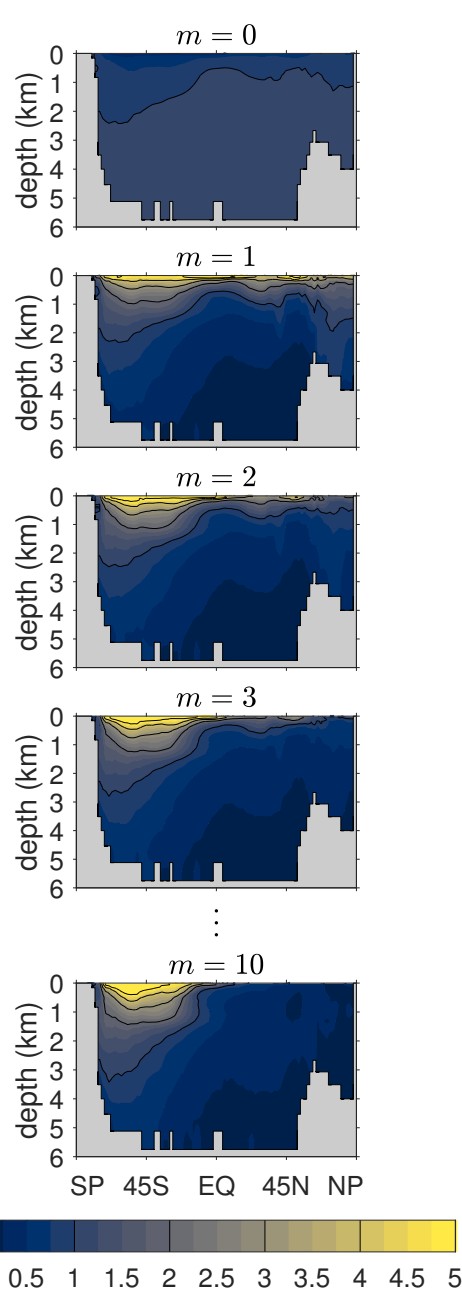

**Figure 8.** Global zonal averages of the fraction of DFe that will be regenerated $m$ times, $f^{m\uparrow}$, normalized by its global mean value, $\langle f^{m\uparrow} \rangle$, for $m = 0, 1, 2, 3$, and 10. This figure is for our typical state.





hydrothermal iron dominates the total DFe inventory, it is possible that one could get $\langle f^{0\downarrow} \rangle_\chi > \langle f^{0\uparrow} \rangle_\chi$ — however, none of our states fits this scenario.

We now return to the asymptotic eigen patterns of $f^{m\uparrow}$. How robust are these eigen patterns across our family of state esti­mates? Figure 9 shows $f^{10\uparrow}$ normalized by its global mean and zonally averaged over the global ocean for our 5 representative
states (for $m = 10$, $f^{m\uparrow}$ is an excellent approximation to the gravest eigenmode of $\widetilde{\mathcal{F}}^{-1}\widetilde{\mathcal{R}}$). While there is some variation across the family, the patterns are qualitatively similar. Because the natural quantity that keeps track of future regenerations is the source-type independent *fraction* of DFe that will undergo $m$ regenerations, the spread in the large-$m$ pattern of $f^{m\uparrow}$ across the family of states is much smaller than the spread in the large-$n$ pattern of $\chi_k^{n\downarrow}$ ($\chi_k^{n\downarrow}$ being the natural quantity for keeping track of past regenerations).

Figure 10a shows the global zonal average of the mean number of future regenerations $\overline{m}(\boldsymbol{r})$, which like $f^{m\uparrow}$ is inde­pendent of source type. The pattern of $\overline{m}$ is similar to the pattern of $f^{1\uparrow}$, which dominates the rapidly converging sum $\overline{m} = \sum_{m=1}^{\infty} m f^{m\uparrow}$. The $\overline{m}$ pattern is surface intensified and largest the Southern Ocean. $\overline{m}$ decays rapidly with depth re­flecting the fact that the deeper the DFe, the harder it will be to escape death on the way to the surface to participate in the biological pump. These qualitative features are robust across the family of state estimates (not shown). Figure 10b shows that
the pattern of the global zonal average of $\overline{m}^{\mathrm{P}}(\boldsymbol{r})$ is almost indistinguishable from that of $\overline{m}(\boldsymbol{r})$. This is because the patterns of $\overline{m}^{\mathrm{P}}$ and $\overline{m}$ are dominated by the phosphorus and iron export productions, respectively, which are similar despite the substantial spatial variations of the Fe:P uptake ratio (Pasquier and Holzer, 2017).

Returning to Figure 7b, we see that both $\langle \overline{m} \rangle$ and $\langle \overline{m}^{\mathrm{P}} \rangle$ are again approximately proportional to $(\sigma_{\mathrm{tot}})^{-1}$, as expected. The magnitude of $\langle \overline{m} \rangle$ remains at or below order unity and ranges from 0.01 to 1.4 as $\sigma_{\mathrm{tot}}$ varies from 41 to $1.9\,\mathrm{Gmol\,yr}^{-1}$.
Correspondingly, $\langle \overline{m}^{\mathrm{P}} \rangle$ ranges from 0.07 to $2.8\,\mathrm{mol\,P\,(mmol\,Fe)}^{-1}$, or from 7 to $290\,\mathrm{mol\,C\,(mmol\,Fe)}^{-1}$ when converted using a C:P ratio of 106:1.

## 5 Natural Iron Fertilization Efficiencies

### 5.1 Export Supported per Unit DFe Injection at $r$

We begin by asking how much source $s_k(\boldsymbol{r})$ contributes to the globally integrated export and remineralization of iron and and
phosphate in organic matter. (In our model all exported phosphorus is instantly respired with the divergence of a Martin POP flux profile – DOP is not explicitly modelled.) We first consider the global export that will be supported per unit injection of DFe at point $\boldsymbol{r}$. The concentration response to a unit injection is the Green function associated with operator $\mathcal{H}$, and the global export due to this response is simply obtained as the global integral of the regeneration operator $\mathcal{R}$ acting on this response. As shown in Appendix E, one obtains that DFe injection into volume element $\mathrm{d}^3\boldsymbol{r}$ at rate $s_k(\boldsymbol{r})\mathrm{d}^3\boldsymbol{r}$ supports a globally integrated
iron export rate, $\Phi_k(\boldsymbol{r})$, given by

$$\Phi_k(\boldsymbol{r}) = \overline{m}(\boldsymbol{r})\, s_k(\boldsymbol{r}) \quad , \tag{13}$$





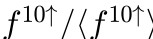

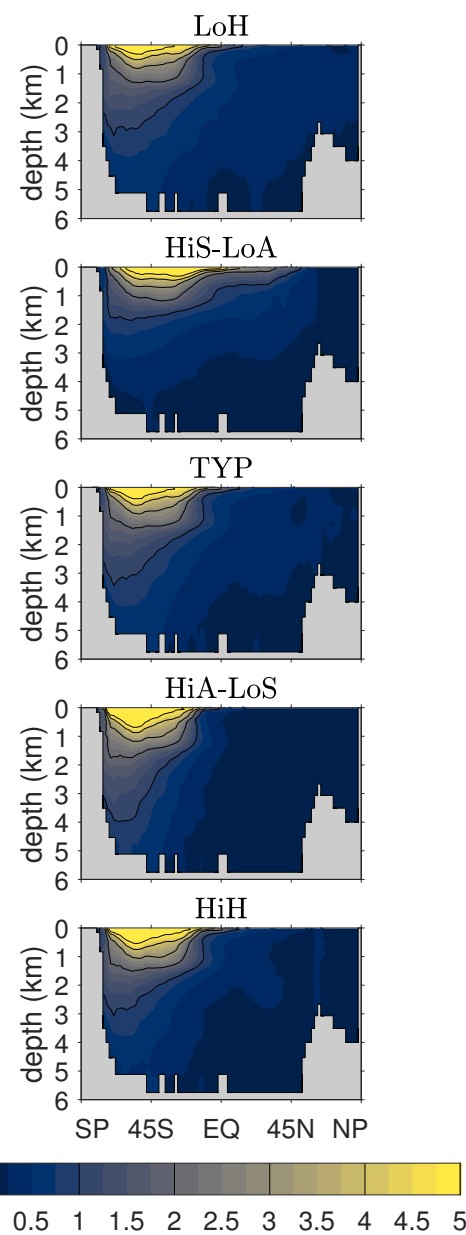

**Figure 9.** Global zonal averages of $f^{10\uparrow}$, normalized by their global means, $\langle f^{10\uparrow} \rangle$, to show the approximate asymptotic eigen patterns for five representative state estimates.




Fertilization efficiency at any point: zonal averages

(a) $[\overline{m}]$

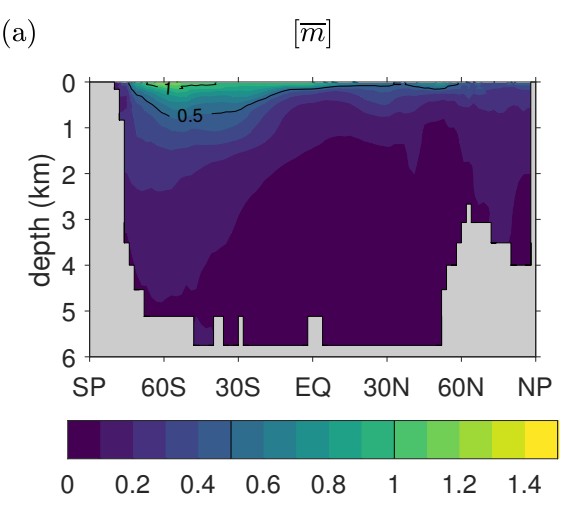

(b) $[\overline{m}^{\mathrm{P}}]$

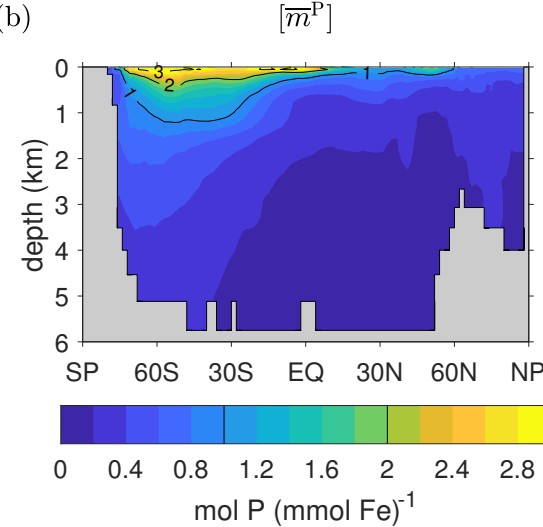

mol P (mmol Fe)$^{-1}$

**Figure 10.** Three-dimensional fertilization efficiency metrics for our typical state, zonally averaged across the global oceans. (a) The mean number of iron molecules, $\overline{m}(\boldsymbol{r})$, that will be regenerated per iron molecule at $\boldsymbol{r}$ during that molecule's lifetime. (b) The corresponding mean number of phosphorus molecules, $\overline{m}^{\mathrm{P}}(\boldsymbol{r})$, that will be regenerated.

while the corresponding phosphorus export, $\Phi_k^{\mathrm{P}}(\boldsymbol{r})$, is given by

$$\Phi_k^{\mathrm{P}}(\boldsymbol{r}) = \overline{m}^{\mathrm{P}}(\boldsymbol{r})\, s_k(\boldsymbol{r}) \quad . \tag{14}$$

These equations have a straightforward interpretation: The mean number of future regenerations, $\overline{m}(\boldsymbol{r})$, is simply the number of DFe molecules that will be exported per DFe molecule injected at point $\boldsymbol{r}$, and $\overline{m}^{\mathrm{P}}(\boldsymbol{r})$ is the corresponding number of phosphorus molecules that will be exported. The quantities $\overline{m}(\boldsymbol{r})$ and $\overline{m}^{\mathrm{P}}(\boldsymbol{r})$ are hence measures of the efficiency of natural
5   iron fertilization at point $\boldsymbol{r}$. Note that this local efficiency is independent of source type; the efficiency is determined by the biological pump and the transport of water, neither of which depend on the iron source type. Indeed, the local efficiency $\overline{m}^{\mathrm{P}}(\boldsymbol{r})$ does not depend on whether there is even a non-zero source at $\boldsymbol{r}$ in the current state of the iron cycle — it quantifies the export that would result *if* DFe were injected at $\boldsymbol{r}$.

Put another way, $\overline{m}(\boldsymbol{r})$ and $\overline{m}^{\mathrm{P}}(\boldsymbol{r})$ quantify the per-molecule fertilization efficiency of the iron that is present at $\boldsymbol{r}$ in the
10   current state of the ocean. These diagnostics are defined without perturbing the system and are hence non-invasive measures of the *natural* iron fertilization efficiency. Thus, the plots of the zonally averaged $\overline{m}(\boldsymbol{r})$ and $\overline{m}^{\mathrm{P}}(\boldsymbol{r})$ in Figure 10 may be interpreted as zonal averages of the three-dimensional iron fertilization efficiency, which can be seen to be highest at the surface and rapidly diminishes with depth. The zonal averages of $\overline{m}$ and $\overline{m}^{\mathrm{P}}$ are dominated by the Southern Ocean. However, we will see that the Southern Ocean only plays a secondary role for the fertilization efficiency at the surface.



Figure 11a shows that the surface patterns of $\overline{m}$ are qualitatively similar across the 5 representative states, with a broad maximum in the central tropical Pacific and with secondary maxima in the subpolar oceans. The states with the largest global mean surface fertilization efficiency, $\langle\overline{m}\rangle_{\mathrm{surf}}$, (values provided on Figure 11a) are the low-source states as one would expect: The less iron is in the system, the more the biological pump is expected to benefit from the iron present. For the HiS-LoA

state, a typical molecule of DFe at the surface leads to a globally integrated export of $\langle\overline{m}\rangle_{\mathrm{surf}} \sim 1.4$ molecules of iron. For the high-source states (HiA-LoS and HiH), both the tropical Pacific and the Southern Ocean are locally most prominent, but the surface mean efficiencies are only $\langle\overline{m}\rangle_{\mathrm{surf}} \sim 0.2$ and $0.5$ molecules of iron exported per typical surface molecule.

Figure 11b shows the surface patterns of the efficiency for fertilizing organic matter export, $\overline{m}^{\mathrm{P}}(\boldsymbol{r})$. The global surface average, $\langle\overline{m}^{\mathrm{P}}\rangle_{\mathrm{surf}}$, is again highest for the low-source states. However, $\langle\overline{m}^{\mathrm{P}}\rangle_{\mathrm{surf}}$ varies less across our representative states

than $\langle\overline{m}\rangle_{\mathrm{surf}}$ (a range of $\sim 3$ compared to $\sim 7$). This is consistent with the fact that all states have very similar phosphorus exports (well constrained by the data used in the optimizations), but widely differing iron exports and Fe:P uptake ratios (Pasquier and Holzer, 2017).

The surface patterns of $\overline{m}^{\mathrm{P}}$ and $\overline{m}$ are similar, with very similar systematic variations across the different states. However, because of the iron dependence of the Fe:P uptake ratio, $\overline{m}^{\mathrm{P}}$ has sharper gradients than $\overline{m}$, with a more pronounced contrast

between the low-efficiency subtropical gyres and the high-efficiency tropical Pacific. The Fe:P uptake ratio in our model is proportional to $\chi/(\chi+k)$, which has its lowest values of around $0.1$ in the central and eastern tropical Pacific for all our states. The corresponding P:Fe uptake ratio (not shown) has a global pattern that is broadly similar to the pattern of the fertilization efficiency $\overline{m}^{\mathrm{P}}$ because large P:Fe means that a relatively large number of P molecules are taken up (and hence exported) per utilized DFe molecule. The correspondence is not exact, but the P:Fe uptake ratio plays a central role in shaping $\overline{m}^{\mathrm{P}}$ at the

surface. At depth, the effect of the P:Fe ratio on $\overline{m}^{\mathrm{P}}$ is less important because, depending on where the DFe re-emerges into the euphotic zone, deep DFe will not necessarily be utilized in regions of high P:Fe.

The key result of Figure 11 is that regardless of state, the tropical Pacific is where iron has its highest natural fertilization efficiency. For high-source states, fertilizing the Southern Ocean can be equally as effective, although the global surface mean efficiency is much lower than for low-source states. The prominence of the tropical Pacific is likely due to the fact that this

region tends to be iron stressed and the equatorial upwelling tends to bring remineralized DFe back to the surface where it is needed to support vigorous biological production. The efficiency is not highest in the eastern tropical Pacific where production is highest because of the associated large scavenging rate due to organic particles. The sweet spot between upwelling-fertilized production and relatively low scavenging lies in the central tropical Pacific to the west of the highest productivity. The state-dependent relative importance of the Southern Ocean compared to the tropical Pacific is a complicated function of how our

optimized states match the nutrient observations and reflects a delicate balance between fertilizing biological production and the resulting enhanced scavenging.

How do our estimates of natural fertilization efficiency compare to previous estimates in the literature? Across our entire family of estimates, $\langle\overline{m}^{\mathrm{P}}\rangle_{\mathrm{surf}}$ ranges from $0.7$ to $7\,\mathrm{mol\,P\,(mmol\,Fe)}^{-1}$, which converts to $73$ to $750\,\mathrm{mol\,C\,(mmol\,Fe)}^{-1}$ using a simple uniform C:P ratio of 106:1. Thus, our estimate of the natural fertilization efficiency is roughly one to two orders

of magnitude larger than the estimate of about $3.3\,\mathrm{mol\,C\,(mmol\,Fe)}^{-1}$ by Zeebe and Archer (2005), who reported $900\,\mathrm{t\,C}$





**Figure 11.** Normalized patterns of our fertilization efficiency metrics at the surface. (a) The mean number of iron molecules, $\overline{m}(\boldsymbol{r})$, that will be regenerated per iron molecule at surface point $\boldsymbol{r}$ during that molecule's lifetime. (b) The corresponding mean number of phosphorus molecules, $\overline{m}^{\mathrm{P}}(\boldsymbol{r})$, that will be regenerated. To allow for a meaningful comparison of the patterns of different states, $\overline{m}(\boldsymbol{r})$ and $\overline{m}^{\mathrm{P}}(\boldsymbol{r})$ have been normalized by their global surface averages, $\langle\overline{m}\rangle_{\mathrm{surf}}$ and $\langle\overline{m}^{\mathrm{P}}\rangle_{\mathrm{surf}}$ (values are given in each plot).

exported per $1.26\,\mathrm{t}$ Fe added. There are at least two reasons why our estimate of fertilization efficiency differs from that of Zeebe and Archer (2005). First, the latter is a regional estimate based on a localized fertilization experiment, while the former



is a global mean estimate. Second, Zeebe and Archer (2005) used the data from the iron fertilization experiments without any corrections for ligand protection in the natural unperturbed system, which leads to an underestimate of the natural fertilization efficiency. de Baar et al. (2008) correct for this in their analysis, giving an estimate of 2.6 to $100\,\mathrm{mol\,C\,(mmol\,Fe)^{-1}}$ for the natural fertilization efficiency. The estimate of de Baar et al. (2008) thus overlaps with the low end of our estimates, which

corresponds to the possibly more realistic high-source states.

## 5.2   Relation to Relative Export Efficiency

To compare the importance of the different iron source types in the current state of the ocean, Pasquier and Holzer (2017) defined the export-support efficiency of iron type $k$ as $\epsilon(s_k) \equiv \widehat{\Phi}_k/\widehat{\sigma}_k$, where $\widehat{\Phi}_k$ is the fraction of the global phosphorus export supported by source type $k$ and $\widehat{\sigma}_k$ is the fractional global source of type $k$. We found that a key metric that is robust

across our family of state estimates is the *relative* export-support efficiency, $e_k^{\mathrm{P}} = \epsilon(s_k)/\epsilon(\widetilde{s}_k)$, which is the ratio of the export-support efficiency of source type $k$ to the export-support efficiency of the other source types, whose combined source is the complement $\widetilde{s}_k \equiv s_{\mathrm{tot}} - s_k$.

It follows algebraically from (14) that the relative export efficiency can be expressed as

$$
e_k^{\mathrm{P}} = \frac{\langle \overline{m}^{\mathrm{P}} \rangle_{s_k}}{\langle \overline{m}^{\mathrm{P}} \rangle_{\widetilde{s}_k}} \quad , \tag{15}
$$

where $\langle \cdot \rangle_{s_k}$ is the $s_k$-weighted global mean, defined so that for any field $x(\boldsymbol{r})$, $\langle x \rangle_{s_k} \equiv \int \mathrm{d}^3\boldsymbol{r}\, x(\boldsymbol{r}) s_k(\boldsymbol{r}) / \int \mathrm{d}^3\boldsymbol{r}\, s_k(\boldsymbol{r})$ and $\langle \cdot \rangle_{\widetilde{s}_k}$

is the corresponding $\widetilde{s}_k$-weighted global mean. Thus, the relative export-support efficiency is the ratio of the mean fertilization efficiency of the source to the mean fertilization efficiency of its complement.

Pasquier and Holzer (2017) found that all members of the family of state estimates share approximately the same relative export efficiencies of $e_{\mathrm{A}}^{\mathrm{P}} = 3.1 \pm 0.8$, $e_{\mathrm{S}}^{\mathrm{P}} = 0.4 \pm 0.2$, and $e_{\mathrm{H}}^{\mathrm{P}} = 0.3 \pm 0.1$, where the uncertainties represent the scatter across the family. What is new here is relationship (15), which relates the global metric $e_k^{\mathrm{P}}$ to source-weighted averages of the local

fertilization efficiency, $\overline{m}^{\mathrm{P}}(\boldsymbol{r})$.

## 6   Discussion and Caveats

Our analysis comes with a number of caveats that should be kept in mind. A number of these were already identified in the work of Pasquier and Holzer (2017), who designed the coupled Fe–P–Si inverse model used here. In particular, the model used DFe data from the GEOTRACES Intermediate Data Product (IDP) 2014 (in addition to an older compilation by Tagliabue

et al. (2012)), which did not contain the newer data from the Pacific and Southern Oceans made available only recently in the GEOTRACES IDP 2017. Although it is inevitable that assimilating the additional constraints of the IDP 2017 would lead to some quantitative changes especially in the Pacific, we think that including the IDP 2017 data would not lead to qualitative changes. The states used here do feature strong Pacific DFe plumes of about the observed spatial extent to the west of the East Pacific Rise (EPR), but unlike the observations these plumes also extend to the east of the EPR. The unrealistic eastward plume

would not be corrected by assimilating the IDP 2017 data into the biogeochemical model because it is due to small biases





in the underlying circulation, which we hold fixed throughout. The deep Pacific circulation can be corrected by assimilating $\delta^3\text{He}$ into a circulation inverse model (*DeVries*, personal communication), but this is beyond the scope of the current study. Moreover, any quantitative differences due to additional constraints would very likely be much smaller than the variations across our family of state estimates, given its two-order-of-magnitude range in iron source strengths.

While the state estimates used were optimized against the observations available at the time, we note that the sedimentary sources of the underlying inverse model of Pasquier and Holzer (2017) are keyed to the flux of particulate organic matter into the bottom box where the organic matter is completely oxidized in our model of the P cycle. This parameterization of the sedimentary source is based on observations along the California coast of a correlation between the DFe flux from sediments and the flux of oxidized organic matter (Elrod et al., 2004). All the model's sediment sources are thus reductive as is the case in

many current iron models (e.g., the PISCES and BEC models, Aumont and Bopp, 2006; Moore and Braucher, 2008). However, the analysis of $\delta^{56}\text{Fe}$ by Conway and John (2014) has highlighted the importance of non-reductive sources, which our inverse model does not include. Our state estimates feature sedimentary sources that are realistically dominated by the continental shelves (Figure H1, Pasquier and Holzer, 2017), but they also include one-to-two orders of magnitude smaller sources below highly productive regions in the eastern tropical Pacific, where a sluggish abyssal circulation allows some DFe to accumulate at

depth, as can be seen in Figure 2. Similarly, our hydrothermal sources inject DFe into the ocean where it is then protected from rapid scavenging by our enhanced ligand concentrations near hydrothermal vents. However, the recent work of Fitzsimmons et al. (2017) suggests that this model of hydrothermal iron needs to be revised in the future to include reversible exchange between DFe and particulate iron, which is currently omitted in our scavenging parametrization.

Other caveats relate to biogeochemical parameters that could not be optimized. The recyclable fraction, $f_{\text{rec}}$, of DFe scav-

enged by opal and POP ($f^{\text{bSi}}$ and $f^{\text{POP}}$ in the notation of Pasquier and Holzer (2017)), was prescribed to be $90\,\%$ for all scavenging particles. $f_{\text{rec}}$ is highly uncertain and in recent models its value has spanned the entire $0$–$100\,\%$ range (e.g., Moore and Braucher, 2008; Galbraith et al., 2010; Frants et al., 2016). We established the sensitivity of our results to the value of $f_{\text{rec}}$ by generating new state estimates from our typical state by prescribing different values of $f_{\text{rec}}$ ranging from $0$ to $100\,\%$ and re-optimizing the scavenging and source parameters following the strategy of Pasquier and Holzer (2017) (see Appendix C for

details). We find that the patterns of $\overline{n}$ and $\overline{m}$ are robust to changes in $f_{\text{rec}}$, although their global mean values, which are order unity for the reoptimized typical state with $f_{\text{rec}} = 0$, systematically decrease by roughly a factor of 2 when $f_{\text{rec}}$ is increased to $100\,\%$. The mean number of past and future phosphorus molecules exported per DFe molecule, $\overline{n}^{\text{P}}$ and $\overline{m}^{\text{P}}$, are robust to changes in $f_{\text{rec}}$ in both pattern and magnitude. This robustness reflects the fact that the optimization can compensate changes in the scavenging pump with changes in the biological iron pump.

The model approximates the iron dependence of the Fe:P uptake ratio by a Monod function with a half saturation constant that is the same for all phytoplankton classes. We acknowledge that this may not be realistic. Furthermore, the model remineralizes iron with the same Fe:P stoichiometric ratio with which it was utilized, i.e., the vertical profiles of iron and phosphate remineralization have identical shapes. However, measurements by Twining et al. (2014) show that, at least for some phytoplankton species, iron is remineralized more slowly than phosphate, suggesting that our remineralization profile for iron could

be too shallow. Because the model is optimized to fit the DFe observations, with an emphasis on deep profiles relative to





surface measurements (Pasquier and Holzer, 2017), a potentially too shallow remineralization of iron would be compensated by an increased strength of the biological pump. Furthermore, the relative amount of scavenging by opal and POP particles is optimizable in our model, so that deeper iron remineralization can be achieved by increasing the scavenging by opal. We acknowledge, however, that when optimizing the match to observed DFe, the model may produce biases in the relative con-
tributions of the biological and scavenging pumps, which would affect our estimates of the number of passages through the biological pump.

The recent work of Rafter et al. (2017) suggests that in surface waters DFe must be preferentially recycled compared to nitrate in order to sustain the observed nitrate consumption in the iron-limited equatorial Pacific. While we do not model the nitrogen cycle, it is reasonable to assume that in the euphotic zone DFe may also be preferentially recycled compared to
phosphate. At face value, this appears to contradict our assumption that the detrital fractions, $f_c$, are identical for Fe, P, and Si export. However, the uptake and export of DFe in our model are proportional to the product $f_c R^{\text{Fe:P}}$, and $R^{\text{Fe:P}}$ is optimized, so that any difference between the iron and phosphate detrital fractions is simply absorbed into $R^{\text{Fe:P}}$. However, this does point to the need for caution when interpreting the optimized values of $R^{\text{Fe:P}}$.

## 7   Conclusions

We have presented a new conceptual and mathematical framework for quantifying the contribution of DFe to the biological pump during its journey from birth by an external source to death by irreversible scavenging and burial. New diagnostics were developed to partition the DFe concentration into the fraction that was regenerated $n$ times in the past or that will be regenerated $m$ times in the future. These diagnostics include new tracer equations for the mean number of regenerations $\overline{n}$ and $\overline{m}$, which afford numerically efficient computation. The mean number of future regenerations that iron at any point $\boldsymbol{r}$ will undergo is a
measure of the fertilization efficiency of iron at $\boldsymbol{r}$, giving the number of iron and phosphorus molecules globally exported per DFe molecule at $\boldsymbol{r}$.

We applied our new diagnostics to a family of optimized state estimates of the global coupled iron, phosphorus, and silicon cycles, assuming steady state. All states of the family match the observed nutrient concentrations about equally well despite spanning a large range of external iron source strengths. Performing our analyses across the family of states allowed us to
identify aspects of the birth-to-death journey of DFe that are robust and to quantify systematic variations with iron source strength. Our key findings are as follows:

1. A large portion of the global DFe inventory never participates in the biological pump. For states with iron sources larger than $\sim 7\,\text{Gmol}\,\text{Fe}\,\text{yr}^{-1}$, more than $50\,\%$ of the inventory has not passed through the pump since birth and more than $85\,\%$ will not pass through the pump before death. Because of its direct injection into the euphotic zone, a larger portion
of aeolian iron passes through the biological pump than other source types. Both the mean number of past and future passages through the biological pump, $\overline{n}$ and $\overline{m}$, are approximately proportional to the bulk iron lifetime $\tau \propto \sigma_{\text{tot}}^{-1}$. For an increase of $\sigma_{\text{tot}}$ from 1.9 to $40\,\text{Gmol}\,\text{Fe}\,\text{yr}^{-1}$, the global average, $\langle \overline{n} \rangle$, decreases from 2.2 to 0.05, while $\langle \overline{m} \rangle$ decreases from 1.4 to 0.01.




2. The three-dimensional distribution of $\overline{n}_k(\boldsymbol{r})$ has its largest values in the Southern Ocean and a pattern that suggests nutrient trapping for all iron source types $k$, but with different vertical structure for different source types. The precise patterns of $\overline{n}_k(\boldsymbol{r})$ are shaped by the delicate balance between regeneration and scavenging, which are processes with strong spatial overlap near the surface. For aeolian iron, the largest values of $\overline{n}_k$ are found in surface, mode, intermediate, and bottom waters, while for sedimentary and hydrothermal iron $\overline{n}_k$ is small in bottom waters because of the greater scavenging hazard during transit from benthic sources to first uptake in the euphotic zone.

3. The spatial distribution of DFe that was regenerated $n$ times since birth varies with $n$ because DFe is reorganized in the water column with each passage through the biological pump. Unused DFe ($n = 0$) is generally concentrated near the sources. The concentration of unused aeolian DFe has a secondary maximum at $\sim 2\,\mathrm{km}$ depth and extends to the sea floor because of the action of the scavenging pump (scavenging and particle transport to depth followed by re-dissolution). Aeolian DFe regenerated exactly once since birth ($n = 1$) has its maximum concentration well below the surface, and sedimentary DFe regenerated exactly once has its maximum concentration well above the bottom, with both maxima at $\sim 1.5\,\mathrm{km}$ depth. DFe trapped in the Southern Ocean has undergone the largest number of regenerations since birth.

4. The pattern of the large-$n$ DFe concentration is independent of iron source type because the memory of birth place quickly dissipates with successive passages through the biological pump, although the rate of convergence to the asymptotic large-$n$ pattern does depend on source type. The large-$n$ pattern corresponds to the gravest eigenmode of $\mathcal{F}^{-1}\mathcal{R}$, which represents the combined transport by the circulation and by sinking particles for one passage through the biological pump. For $n$ larger than $\sim 5$, the concentration of DFe that was regenerated $n$ times since birth decays exponentially with $n$. The $e$-folding scale $n^*$ ranges from $0.4$–$2.8$ across our family, with higher sources corresponding to smaller $n^*$ (faster decay consistent with shorter lifetimes).

5. The fraction $f^{m\uparrow}(\boldsymbol{r})$ of DFe currently at $\boldsymbol{r}$ that will pass $m$ times through the biological pump in the future is independent of iron source type. The local fraction that will not participate in biological production in the future, $f^{0\uparrow}(\boldsymbol{r})$, has a nearly spatially uniform distribution and makes the largest contribution to the DFe inventory. The local fraction that will be utilized exactly once before death, $f^{1\uparrow}(\boldsymbol{r})$, is typically concentrated near the surface, where the likelihood of passing through the biological pump before being scavenged is largest. Correspondingly, the three-dimensional distribution of $\overline{m}(\boldsymbol{r})$ has its maximum near the surface and is Southern Ocean intensified.

6. The fraction of DFe that will be regenerated $m$ times in the future again approaches an eigenmode for asymptotically large $m$. The amplitude of this eigenmode decays with the same $e$-folding scale $n^*$ as the eigenmode associated with past regenerations. However, the eigenmodes associated with future and past regenerations have different patterns, with the mode for future regenerations being much more surface intensified. Both past and future modes are concentrated in the Southern Ocean because multiple regenerations during an iron molecule's lifetime are more likely where nutrients are effectively trapped (Primeau et al., 2013; Holzer et al., 2014).



7. We defined and quantified the natural iron fertilization efficiency at an arbitrary point $r$ in terms of the number of globally exported phosphorus molecules per DFe molecule at $r$. This is the first time that the iron fertilization efficiency has been estimated in three dimensions, which may ultimately prove useful for quantifying the importance of different iron reservoirs for supporting the ocean's global export production. At the surface, the natural fertilization efficiency is largest in the central tropical Pacific with secondary maxima in the subpolar oceans. The relative importance of the Southern Ocean is greatest for high-source states. Globally averaged, the surface fertilization efficiency ranges from $0.7$ to $7\,\mathrm{mol\,P\,(mmol\,Fe)^{-1}}$ across our family of state estimates, with the low-source states having the largest efficiencies. (In carbon units, the efficiency ranges from $73$ to $750\,\mathrm{mol\,C\,(mmol\,Fe)^{-1}}$ using a C:P ratio of 106:1.)

Intimately connected to the mean number of past and future regenerations are the age and expected remaining lifetimes of DFe in the ocean. A full exploration of these timescales and the associated transit-time distributions is beyond the scope of this study. However, in future work, we plan to explore the timescales of the iron cycle and their connection to setting the efficiency with which iron fertilization achieves carbon sequestration (e.g., DeVries et al., 2012; Pasquier and Holzer, 2016). Finally, the concepts and methods employed here can be applied to other nutrients for a more complete picture of how the interaction between the biological pump and the physical transport shapes their distributions and cycling rates; we plan to do so in future work.

## Appendix A: Local Iron Death Rate

To diagnose the local death rate with which iron is permanently removed from the ocean, we first rewrite the reversible scavenging operator $\mathcal{D}$ in terms of a simplified local specific death rate. The operator $\mathcal{D}$ is an integral operator so that $(\mathcal{D}\chi)(r) = \int\mathrm{d}^3r'\,K_{\mathcal{D}}(r|r')\chi(r')$ with adjoint $(\widetilde{\mathcal{D}}\phi)(r') = \int\mathrm{d}^3r\,\phi(r')K_{\mathcal{D}}(r|r')$. The term $K_{\mathcal{D}}(r|r')\chi(r')$ represents the rate with which iron is scavenged at $r'$ minus the rate with which this scavenged iron is redistributed to points $r$ through particle transport and re-dissolution. By integrating over all destination points $r$, we obtain the deficit between the rate of iron scavenging at $r'$ and the total water-column integrated rate of redissolution of that iron. Thus, $d(r') \equiv \int\mathrm{d}^3r\,K_{\mathcal{D}}(r|r')\chi(r')$ is the rate of permanent removal or death at $r'$. In terms of the equivalent linear specific death rate, we have $d(r') = \gamma^{\mathcal{D}}(r')\chi(r')$, with $\gamma^{\mathcal{D}}(r') \equiv \int\mathrm{d}^3r\,K_{\mathcal{D}}(r|r')$.

For the interpretation of our diagnostics it is helpful to know how the local iron death rate, $d(r') = \gamma^{\mathcal{D}}(r')\chi(r')$, is distributed through the ocean. Figure A1 shows basin and global zonal averages of the death rate for our five representative states. Key features common to all states are relatively low death rates in the subtropical gyres of both hemispheres where production and hence scavenging particle fluxes are low, and relatively high death rates at high latitudes and in the tropics where production is large. The highest death rates occur in the surface ocean where particle fluxes are highest. The near-surface death rates are larger in the Northern Hemisphere than in the Southern Hemisphere, presumably because the DFe concentrations are higher in the Northern Hemisphere.





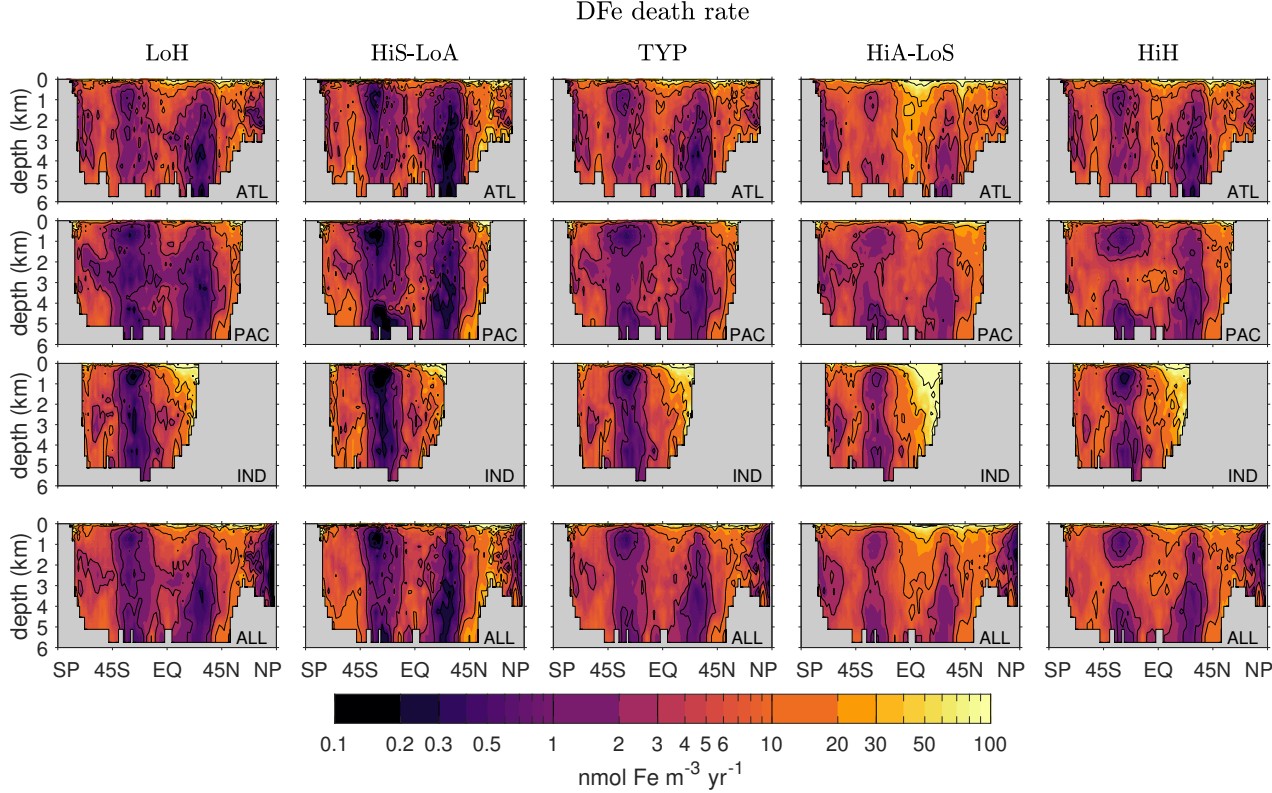

**Figure A1.** Basin and global zonal averages of the death rate $d(\boldsymbol{r'})$ for our five representative states. Note the logarithmic color scale.

## Appendix B: Derivation of Equation for $\overline{n}(\boldsymbol{r})$

First we verify that $\sum_{n=0}^{\infty} \chi_k^{n\downarrow} = \chi_k$. From (4) and (5) we have

$$\chi_k^{n\downarrow} = \mathcal{A}^n \mathcal{F}^{-1} s_k \quad , \tag{B1}$$

where $\mathcal{A} \equiv \mathcal{F}^{-1}\mathcal{R}$. Because $\mathcal{A}$ must have a maximum eigenvalue less than unity for convergence, we can use the geometric operator sum $\sum_{n=0}^{\infty} \mathcal{A}^n = (1-\mathcal{A})^{-1}$. Thus,

$$\sum_{n=0}^{\infty} \chi_k^{n\downarrow} = (1-\mathcal{A})^{-1} \mathcal{F}^{-1} s_k \quad . \tag{B2}$$

5   Applying $\mathcal{F}(1-\mathcal{A})$ from the left, and recognizing that $\mathcal{F}(1-\mathcal{A}) = \mathcal{F} - \mathcal{R} = \mathcal{H}$, we have

$$\mathcal{H} \sum_{n=0}^{\infty} \chi_k^{n\downarrow} = s_k \quad . \tag{B3}$$

This shows that $\sum_{n=0}^{\infty} \chi_k^{n\downarrow}$ and $\chi_k$ obey the same equation and hence that they are equal.





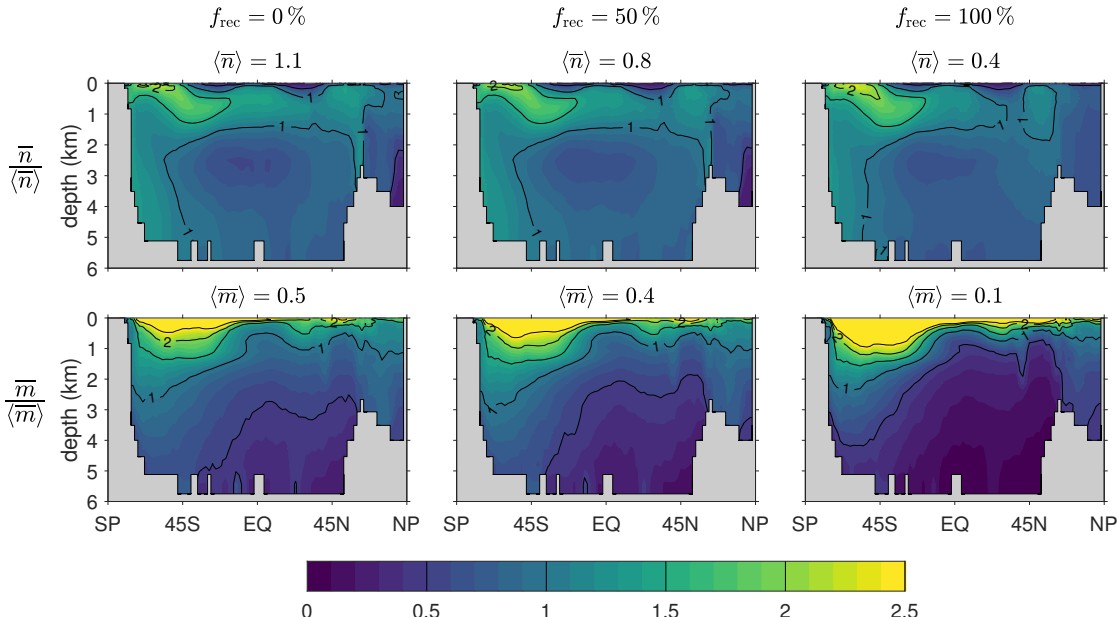

**Figure C1.** Top: The zonal means of $\overline{n}(\boldsymbol{r})$, normalized by the corresponding global averages $\langle \overline{n} \rangle$, for our typical state estimate re-optimized with three very different recyclable fractions: $f_{\text{rec}} = 0, 50$, and $100\,\%$. Bottom: The corresponding normalized zonal means of $\overline{m}(\boldsymbol{r})$. The values of the global averages, $\langle \overline{n} \rangle$ and $\langle \overline{m} \rangle$, are given in the plot titles.

The defining equation for $\overline{n}_k(\boldsymbol{r})$ is

$$\chi_k(\boldsymbol{r})\overline{n}_k(\boldsymbol{r}) = \sum_{n=0}^{\infty} n\chi_k^{n\downarrow}(\boldsymbol{r}) \quad . \tag{B4}$$

From recursion relation (5), we have $\chi_k^{n\downarrow} = \mathcal{A}^n \chi_k^{0\downarrow}$. Substituting this, using the geometric operator sum $\sum_{n=0}^{\infty} n\mathcal{A}^n = (1 - \mathcal{A})^{-1}\mathcal{A}(1 - \mathcal{A})^{-1}$, and the fact that from (B1) $\chi_k^{0\downarrow} = \mathcal{F}^{-1}s_k = \mathcal{F}^{-1}\mathcal{H}\mathcal{H}^{-1}s_k = (1 - \mathcal{A})\chi_k$, we obtain

$$\chi_k \overline{n}_k = (1 - \mathcal{A})^{-1}\mathcal{A}\chi_k \quad . \tag{B5}$$

Applying $\mathcal{R}\mathcal{A}^{-1}(1 - \mathcal{A}) = \mathcal{F} - \mathcal{R} = \mathcal{H}$ from the left yields (6).

## 5 Appendix C: Variation of Key Diagnostics with Recyclable Fraction Parameter

To explore variations of our results with the non-optimized recyclable fraction $f_{\text{rec}}$, we changed the value of $f_{\text{rec}}$ of our typical state (for which $f_{\text{rec}} = 90\,\%$), and then re-optimized the biogeochemical sink and source parameters following the optimization strategy described by Pasquier and Holzer (2017). This generated a set of 8 optimized state estimates with $f_{\text{rec}} = 0, 10, 30, 50, 70, 80, 90$, and $100\,\%$. Except for the $f_{\text{rec}} = 100\,\%$ case, these states have similar iron source strengths ($\sigma_{\text{A}}$ ranges from 5.3 to $5.7\,\text{Gmol yr}^{-1}$, $\sigma_{\text{S}}$ from 1.7 to $1.9\,\text{Gmol yr}^{-1}$, and $\sigma_{\text{H}}$ is unchanged to 2 significant figures at




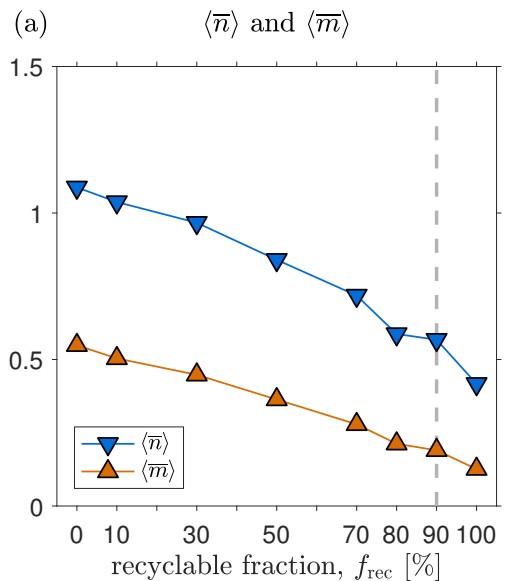
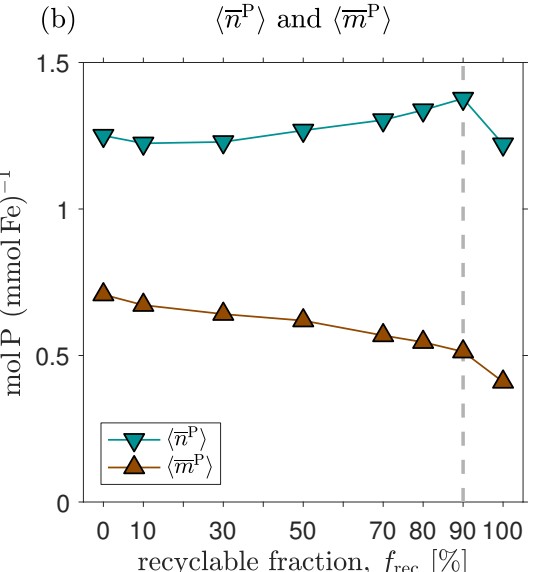

**Figure C2.** (a) The globally averaged mean number of past and future DFe regenerations per injected DFe molecule, $\langle \overline{n} \rangle$ and $\langle \overline{m} \rangle$, as a function of the recyclable fraction $f_{\mathrm{rec}}$. The corresponding states were obtained by changing the value of $f_{\mathrm{rec}}$ of the typical state and re-optimizing all other source and scavenging parameters. The typical state has $f_{\mathrm{rec}} = 90\,\%$ and is indicated by the vertical dashed line. (b) As (a) for the past and future number of phosphorus molecules exported per injected DFe molecule, $\langle \overline{n}^{\mathrm{P}} \rangle$ and $\langle \overline{m}^{\mathrm{P}} \rangle$.

$0.88\,\mathrm{Gmol\,yr^{-1}}$). For the extreme case of $f_{\mathrm{rec}} = 100\,\%$, when the only permanent iron sink in the system is due to the flux of scavenged iron that reaches the ocean bottom where we assume it is buried (in addition, iron scavenged by mineral dust is always assumed not to be recyclable, but dust scavenging is very small for our estimates), the optimized sedimentary iron source more than doubles and the other sources also undergo significant adjustments during the re-optimization ($\sigma_{\mathrm{A}} = 5.1$,

$\sigma_{\mathrm{S}} = 3.7$, and $\sigma_{\mathrm{H}} = 0.85\,\mathrm{Gmol\,yr^{-1}}$ for $f_{\mathrm{rec}} = 100\,\%$).

Figure C1 shows that the spatial pattern of the zonally averaged $\overline{n}$ is very insensitive to the value of $f_{\mathrm{rec}}$ when the scavenging and source parameters are optimized. The patterns of the zonally averaged $\overline{m}$ have the same qualitative features for all values of $f_{\mathrm{rec}}$, but become more surface-intensified as $f_{\mathrm{rec}}$ increases to unity, presumably because a greater recycling fraction increases the chance that DFe is available near the euphotic zone to pass through the biological pump. The patterns of $\overline{n}^{\mathrm{P}}$ and $\overline{m}^{\mathrm{P}}$ (not

shown) are similarly insensitive to $f_{\mathrm{rec}}$.

Figure C2 shows how the volume-weighted global means of $\overline{n}, \overline{m}, \overline{n}^{\mathrm{P}}$, and $\overline{m}^{\mathrm{P}}$ vary with the recyclable fraction, $f_{\mathrm{rec}}$. We find that $\langle \overline{n} \rangle$ and $\langle \overline{m} \rangle$ are order unity for $f_{\mathrm{rec}} = 0\,\%$ and tend to decrease by roughly a factor of 2 to 3 when $f_{\mathrm{rec}}$ is increased from 0 to $100\,\%$. We emphasize that this is not simple sensitivity to $f_{\mathrm{rec}}$, because the scavenging and source parameters have been reoptimized for each choice of $f_{\mathrm{rec}}$. Broadly, a larger value of the number of regenerations means that more iron is transported

to depth with the biological pump. This is consistent with the fact that for small values of $f_{\mathrm{rec}}$, less iron is transported to depth through scavenging and re-dissolution. Given the similar iron sources of these states, and hence similar global mean scavenging





rates, reduced iron transport through scavenging must be compensated by increased transport through biological uptake and regeneration, as measured here by the mean number or iron regenerations. The mean number of phosphorus molecules that are regenerated in the past or the future per iron molecule, $\overline{n}^{\mathrm{P}}$ and $\overline{m}^{\mathrm{P}}$, are much less sensitive to the value of $f_{\mathrm{rec}}$ in these optimized states. The reason for this insensitivity probably lies in the fact that the phosphorus export is well constrained by the

nutrient and phytoplankton data against which we optimize (Pasquier and Holzer, 2017), and any changes in iron export are compensated by different optimized values of the Fe:P uptake ratio, which we also use as the Fe:P ratio for remineralization.

## Appendix D: Recursion Relation for $f^{m\uparrow}$

### D1   Green Functions

To derive the recursion relation for the fraction of iron at a given location that is regenerated exactly $m$ times before death, we

use a Green function approach. The Green function associated with the equation $(\partial_t + \mathcal{H})\chi = s$ is obtained by replacing the source $s$ with a Dirac delta function in space and time:

$$(\partial_t + \mathcal{H}_{\boldsymbol{r}})G_{\mathcal{H}}(\boldsymbol{r},t|\boldsymbol{r}',t') = \delta(t-t')\delta^3(\boldsymbol{r}-\boldsymbol{r}') \quad , \tag{D1}$$

where the subscript on $\mathcal{H}_{\boldsymbol{r}}$ reminds us that $\mathcal{H}$ acts on the field-point coordinates $\boldsymbol{r}$. The Green function for the time-reversed adjoint flow (e.g., Holzer and Hall, 2000) obeys

$$(-\partial_{t'} + \widetilde{\mathcal{H}}_{\boldsymbol{r}'})G_{\widetilde{\mathcal{H}}}(\boldsymbol{r}',t'|\boldsymbol{r},t) = \delta(t-t')\delta^3(\boldsymbol{r}-\boldsymbol{r}') \quad , \tag{D2}$$

where the adjoint Green function $G_{\widetilde{\mathcal{H}}}$ obeys the reciprocity relation $G_{\widetilde{\mathcal{H}}}(\boldsymbol{r}',t'|\boldsymbol{r},t) = G_{\mathcal{H}}(\boldsymbol{r},t|\boldsymbol{r}',t')$. The Green functions

associated with $(\partial_t + \mathcal{F})\chi = s$ are defined in exactly the same manner.

    All adjoints here are defined in terms of the volume-weighted inner product so that for linear operator $\mathcal{H}$ and any two fields $\phi$ and $\psi$ we have $\int \mathrm{d}^3\boldsymbol{r}\,\phi(\boldsymbol{r})(\mathcal{H}\psi)(\boldsymbol{r}) = \int \mathrm{d}^3\boldsymbol{r}\,(\widetilde{\mathcal{H}}\phi)(\boldsymbol{r})\psi(\boldsymbol{r})$. For computation, all linear operators are discretized on a numerical grid and organized into sparse matrices, e.g., $\mathcal{H}$ becomes matrix $\mathbf{H}$ with adjoint $\widetilde{\mathbf{H}} = \mathbf{V}^{-1}\mathbf{H}^{\mathsf{T}}\mathbf{V}$, where $\mathbf{V}$ is a diagonal matrix of the grid box volumes.

### 20   D2   All DFe Must Die

Now consider the concentration of iron $\chi(\boldsymbol{r}',t')$ in some volume $\mathrm{d}^3\boldsymbol{r}'$. As the system evolves the mass $\chi(\boldsymbol{r}',t')\,\mathrm{d}^3\boldsymbol{r}'$ results in concentration $\mathcal{X}(\boldsymbol{r},t|\boldsymbol{r}',t')\,\mathrm{d}^3\boldsymbol{r}'$ at $(\boldsymbol{r},t)$ which is obtained by propagating with $G_{\mathcal{H}}$ so that

$$\mathcal{X}(\boldsymbol{r},t|\boldsymbol{r}',t')\,\mathrm{d}^3\boldsymbol{r}' = G_{\mathcal{H}}(\boldsymbol{r},t|\boldsymbol{r}',t')\chi(\boldsymbol{r}',t')\,\mathrm{d}^3\boldsymbol{r}' \quad . \tag{D3}$$

(Note that there is no integral as this is pointwise propagation.) The death rate per unit volume incurred by $\mathcal{X}(\boldsymbol{r},t|\boldsymbol{r}',t')\,\mathrm{d}^3\boldsymbol{r}'$ at $(\boldsymbol{r},t)$ is given by $\gamma^{\mathcal{D}}(\boldsymbol{r})\mathcal{X}(\boldsymbol{r},t|\boldsymbol{r}',t')\,\mathrm{d}^3\boldsymbol{r}'$. Integrating this death rate over all times $t$ and over the entire ocean for $\boldsymbol{r}$, we must

recover the initial mass $\chi(\boldsymbol{r}',t')\,\mathrm{d}^3\boldsymbol{r}'$. Thus, we have

$$\chi(\boldsymbol{r}',t')\,\mathrm{d}^3\boldsymbol{r}' \equiv \int \mathrm{d}t \int \mathrm{d}^3\boldsymbol{r}\,\gamma^{\mathcal{D}}(\boldsymbol{r})\,G_{\mathcal{H}}(\boldsymbol{r},t|\boldsymbol{r}',t')\,\chi(\boldsymbol{r}',t')\,\mathrm{d}^3\boldsymbol{r}' \quad , \tag{D4}$$





where the initial volume $\mathrm{d}^3\boldsymbol{r}'$ is *not* integrated over. Because $G_{\mathcal{H}}$ is just a function and not a differential operator, and because we are integrating with respect to $(\boldsymbol{r}, t)$, we can divide both sides by $\chi(\boldsymbol{r}', t')\,\mathrm{d}^3\boldsymbol{r}'$, which gives

$$f^{\uparrow} \equiv \int \mathrm{d}t \int \mathrm{d}^3\boldsymbol{r}\, \gamma^{\mathcal{D}}(\boldsymbol{r}) G_{\mathcal{H}}(\boldsymbol{r}, t | \boldsymbol{r}', t') \quad , \tag{D5}$$

where $f^{\uparrow}$ is just a spatially uniform field of unit value, i.e., $f^{\uparrow} = 1$. Applying the time-reversed adjoint operator $(-\partial_{t'} + \widetilde{\mathcal{H}}_{\boldsymbol{r}'})$ from the left to (D5) and using the reciprocity relation of $G_{\mathcal{H}}$, we obtain (8). Note that $\widetilde{\mathcal{H}} = \widetilde{\mathcal{T}} + (\widetilde{\mathcal{L}} - \widetilde{\mathcal{R}}) + \widetilde{\mathcal{D}}$. Because $\mathcal{T}$ and

$\mathcal{L} - \mathcal{R}$ are mass-conserving operators, $\widetilde{\mathcal{T}} f^{\uparrow} = 0$ and $(\widetilde{\mathcal{L}} - \widetilde{\mathcal{R}}) f^{\uparrow} = 0$ so that (8) is equivalent to $\widetilde{\mathcal{D}} f^{\uparrow} = \gamma^{\mathcal{D}}$, which reproduces the definition of $\gamma^{\mathcal{D}}$.

**D3   Fraction Not Regenerated in the Future**

Similarly we can construct the concentration of DFe that has been regenerated exactly zero times since being at $(\boldsymbol{r}', t')$. By propagating the mass $\chi(\boldsymbol{r}', t')\,\mathrm{d}^3\boldsymbol{r}'$ with $\mathcal{F}$ instead of $\mathcal{H}$, we obtain the resulting concentration at $t$ that has not passed through

the biological pump since $t'$:

$$\mathcal{X}^{0\uparrow}(\boldsymbol{r}, t | \boldsymbol{r}', t')\,\mathrm{d}^3\boldsymbol{r}' = G_{\mathcal{F}}(\boldsymbol{r}, t | \boldsymbol{r}', t')\chi(\boldsymbol{r}', t')\mathrm{d}^3\boldsymbol{r}' \quad . \tag{D6}$$

Calculating the death rate per unit volume by multiplying with $\gamma^{\mathcal{D}}$ and integrating over all $\boldsymbol{r}$ and $t$ must give the mass in $\mathrm{d}^3\boldsymbol{r}'$ that will not be regenerated in the future, i.e., $\chi^{0\uparrow}(\boldsymbol{r}', t')\,\mathrm{d}^3\boldsymbol{r}'$. Dividing both sides by the starting mass $\chi(\boldsymbol{r}', t')\,\mathrm{d}^3\boldsymbol{r}'$ and defining the fraction $f^{0\uparrow}(\boldsymbol{r}', t') \equiv \chi^{0\uparrow}(\boldsymbol{r}', t')/\chi(\boldsymbol{r}', t')$ gives

$$f^{0\uparrow}(\boldsymbol{r}', t') = \int \mathrm{d}t \int \mathrm{d}^3\boldsymbol{r}\, \gamma^{\mathcal{D}}(\boldsymbol{r}) G_{\mathcal{F}}(\boldsymbol{r}, t | \boldsymbol{r}', t') \quad . \tag{D7}$$

Applying the time-reversed adjoint operator $(\partial_{t'} + \widetilde{\mathcal{F}}_{\boldsymbol{r}'})$ from the left gives (9).

**D4   Recursion for the Fraction Regenerated $m$ times in the Future**

To construct the recursion equation for $f^{m\uparrow}$, it is useful to explicitly write the regeneration operator in terms of its integration kernel

$$(\mathcal{R}\chi)(\boldsymbol{r}) = \int \mathrm{d}^3\boldsymbol{r}''\, K_{\mathcal{R}}(\boldsymbol{r}|\boldsymbol{r}'')\chi(\boldsymbol{r}'', t') \quad . \tag{D8}$$

We again consider the concentration at $(\boldsymbol{r}, t)$ resulting from the mass of DFe in $\mathrm{d}^3\boldsymbol{r}'$ at $t'$ that was not regenerated since $t'$, $\mathcal{X}^{0\uparrow}(\boldsymbol{r}, t | \boldsymbol{r}', t')\,\mathrm{d}^3\boldsymbol{r}'$, as defined by (D6). The rate with which $\mathcal{X}^{0\uparrow}(\boldsymbol{r}, t | \boldsymbol{r}', t')\,\mathrm{d}^3\boldsymbol{r}'$ is regenerated (i.e., remineralized) in volume

$\mathrm{d}^3\boldsymbol{r}''$ is given by $\int \mathrm{d}^3\boldsymbol{r}\, K_{\mathcal{R}}(\boldsymbol{r}''|\boldsymbol{r})\mathcal{X}^{0\uparrow}(\boldsymbol{r}, t | \boldsymbol{r}', t')\,\mathrm{d}^3\boldsymbol{r}'$ (where $\mathrm{d}^3\boldsymbol{r}'$ is not integrated over) and the fraction of this rate that will get regenerated a further $m$ times in the future is given by

$$f^{m\uparrow}(\boldsymbol{r}'') \int \mathrm{d}^3\boldsymbol{r}\, K_{\mathcal{R}}(\boldsymbol{r}''|\boldsymbol{r})G_{\mathcal{F}}(\boldsymbol{r}, t | \boldsymbol{r}', t')\chi(\boldsymbol{r}', t')\mathrm{d}^3\boldsymbol{r}' \quad . \tag{D9}$$



Integrating this rate over all $\boldsymbol{r}''$ and $t$, we must recover the fraction of the initial mass at $t'$ that will be regenerated exactly $m+1$ times in the future. Hence we have

$$f^{(m+1)\uparrow}(\boldsymbol{r}')\chi(\boldsymbol{r}',t')\,\mathrm{d}^3\boldsymbol{r}' = \int\mathrm{d}t\int\mathrm{d}^3\boldsymbol{r}\int\mathrm{d}^3\boldsymbol{r}''\,f^{m\uparrow}(\boldsymbol{r}'')$$
$$\times K_{\mathcal{R}}(\boldsymbol{r}''|\boldsymbol{r})G_{\mathcal{F}}(\boldsymbol{r},t|\boldsymbol{r}',t')\chi(\boldsymbol{r}',t')\mathrm{d}^3\boldsymbol{r}' \quad . \quad \text{(D10)}$$

Dividing both sides by the initial mass $\chi(\boldsymbol{r}',t')\,\mathrm{d}^3\boldsymbol{r}'$ and with the adjoint regeneration operator $\widetilde{\mathcal{R}}$ defined through $(\widetilde{\mathcal{R}}f)(\boldsymbol{r}) = \int\mathrm{d}^3\boldsymbol{r}''\,f(\boldsymbol{r}'')K_{\widetilde{\mathcal{R}}}(\boldsymbol{r}|\boldsymbol{r}'')$, where $K_{\widetilde{\mathcal{R}}}(\boldsymbol{r}|\boldsymbol{r}'') \equiv K_{\mathcal{R}}(\boldsymbol{r}''|\boldsymbol{r})$, we have

$$f^{(m+1)\uparrow}(\boldsymbol{r}') = \int\mathrm{d}t\int\mathrm{d}^3\boldsymbol{r}''\int\mathrm{d}^3\boldsymbol{r}\,G_{\widetilde{\mathcal{F}}}(\boldsymbol{r}',t'|\boldsymbol{r},t)$$
$$\times K_{\widetilde{\mathcal{R}}}(\boldsymbol{r}|\boldsymbol{r}'')f^{m\uparrow}(\boldsymbol{r}'') \quad . \quad \text{(D11)}$$

Note that (D11) propagates the fraction to be regenerated $m$ times through one regeneration (modelled as instantaneous) and through the adjoint flow backward in time to the fraction that will be regenerated $(m+1)$ times. Applying $(-\partial_{t'} + \widetilde{\mathcal{F}})$ from the left gives the recursion equation (10). Note that this recursion equation can be written as $f^{m\uparrow} = (\widetilde{\mathcal{F}}^{-1}\widetilde{\mathcal{R}})^m f^{0\uparrow}$. Using (9) and regrouping the factors of $\widetilde{\mathcal{F}}^{-1}\widetilde{\mathcal{R}}$, this recursion relation can be re-written as

$$f^{m\uparrow} = \widetilde{\mathcal{F}}^{-1}\widetilde{\mathcal{A}}^m\gamma^{\mathcal{D}} \quad , \tag{D12}$$

where $\mathcal{A} \equiv \mathcal{F}^{-1}\mathcal{R}$ as in Appendix B, with adjoint $\widetilde{\mathcal{A}} = \widetilde{\mathcal{R}}\widetilde{\mathcal{F}}^{-1}$. Equation (D12) is the analog of Equation (B1) for the time-
reversed adjoint problem.

    Using similar techniques as in Appendix B, it follows readily that $\sum_{m=0}^{\infty} f^{m\uparrow} = f^{\uparrow} = 1$, as must be the case. Equation (11) for $\overline{m}$ is also derived analogously to $\overline{n}$ in Appendix B.

## Appendix E: Relation Between Export and $\overline{m}$

Here we calculate the steady-state globally integrated export that results from a steady injection with source $s(\boldsymbol{r}')$ (in $\mathrm{mol\,Fe\,m^{-3}\,s^{-1}}$)
in some volume $\mathrm{d}^3\boldsymbol{r}'$ during $\mathrm{d}t'$. Propagating the initial DFe mass $s(\boldsymbol{r}')\,\mathrm{d}^3\boldsymbol{r}'\,\mathrm{d}t'$ with $\mathcal{H}$ results in concentration $G_{\mathcal{H}}(\boldsymbol{r},t|\boldsymbol{r}',t')s(\boldsymbol{r}')\,\mathrm{d}^3\boldsymbol{r}'\,\mathrm{d}t'$ at $(\boldsymbol{r},t)$. This DFe is instantly regenerated at $\boldsymbol{r}''$ with rate $\int\mathrm{d}^3\boldsymbol{r}\,K_{\mathcal{R}}(\boldsymbol{r}''|\boldsymbol{r})G_{\mathcal{H}}(\boldsymbol{r},t|\boldsymbol{r}',t')s(\boldsymbol{r}')\,\mathrm{d}^3\boldsymbol{r}'\,\mathrm{d}t'$. Integrating over all $\boldsymbol{r}''$ and $t$ must give the globally integrated export $\Phi(\boldsymbol{r}')\,\mathrm{d}^3\boldsymbol{r}'\,\mathrm{d}t'$ that is due to the initial injection of the mass $s(\boldsymbol{r}')\,\mathrm{d}^3\boldsymbol{r}'\,\mathrm{d}t'$, and further dividing by $\mathrm{d}^3\boldsymbol{r}'\,\mathrm{d}t'$ gives

$$\Phi(\boldsymbol{r}') = \int\mathrm{d}^3\boldsymbol{r}''\int\mathrm{d}^3\boldsymbol{r}\,K_{\mathcal{R}}(\boldsymbol{r}''|\boldsymbol{r})\int\mathrm{d}t\,G_{\mathcal{H}}(\boldsymbol{r},t|\boldsymbol{r}',t')s(\boldsymbol{r}') \quad . \tag{E1}$$

The globally integrated export per unit source injection at $\boldsymbol{r}'$ is thus $g(\boldsymbol{r}') \equiv \Phi(\boldsymbol{r}')/s(\boldsymbol{r}')$. Dividing (E1) by $s(\boldsymbol{r}')$ and applying
$(-\partial_{t'} + \widetilde{\mathcal{H}})$ gives

$$\widetilde{\mathcal{H}}g = \widetilde{\mathcal{R}}f^{\uparrow} \quad , \tag{E2}$$





which is the Green function that propagates a unit source to globally integrated export. Comparison with (11) shows that $g = \overline{m}$ from which (13) of the main text follow by replacing our generic source $s(\boldsymbol{r}')$ with one of the sources $s_k(\boldsymbol{r}')$. Equation (14) for the global phosphorus export due to DFe injection at $\boldsymbol{r}'$ can be derived in exactly the same way by using regeneration operator $\mathcal{R}^{\mathrm{P}}$ instead of $\mathcal{R}$.

5  *Acknowledgements.* Benoît Pasquier acknowledges support from the Government of Monaco, the Scientific Centre of Monaco, the Frères Louis et Max Principale Foundation, the Cuomo Foundation, J. K. Moore (DOE Grant DE-SC0016539 and NSF Grant 1658380) and F. Primeau (NSF Grant 1658380). Mark Holzer acknowledges a UNSW Goldstar award. The state estimates of the iron cycle used here are available from the authors on request.




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
