# Peer review of "The number of past and future regenerations of iron in the ocean and its intrinsic fertilization efficiency"

_Biogeosciences, 2018_

## Referee Comment (RC1) · Anonymous Referee #1 · 7 Sep 2018

General Comments: This is a review of "Iron fertilization efficiency and the number of past and future regenerations of iron in the ocean" by B. Pasquier and M. Holzer. The authors present a novel technique to track the life-cycle of dissolved iron (DFe) in the ocean. The authors apply the technique to multiple instances of data-constrained representations of the iron cycle and investigate how many cycles DFe parcels experience before and after participating in the biological pump. The authors use their technique to quantify the efficiency of iron fertilization on biological export, one of the motivations for their study. The manuscript is fairly well written, laying out clearly their novel technique, and they present well-designed experiments to utilize their technique. I think this manuscript will be a useful addition to the literature. I am making suggestions for some

minor changes and/or additions.

Specific Comments:

The abstract doesn't mention that the model used by the authors includes multiple types of external sources of DFe. I think it would be useful to mention in the abstract the types of external sources of DFe under consideration.

The notation for the nonlinear model in Section 2.1 deviates from the notation in the author's previous work (Pasquier & Holzer, Biogeosciences, 2017) (e.g., removal of the Redfield ratio for the uptake terms, and changed notation for the particle transport terms). I suggest that the authors either use the same notation as their previous work, or describe how and why the notation in the current work differs from the previous work.

There is no motivation given for the definition of the equivalent linear model in Section 2.2. I think the paper would benefit from having a paragraph describing what the goals/requirements of the equivalent linear model are, and how goals/requirements lead to the model that the authors are using.

While some symbols chosen for the various terms in the nonlinear and equivalent linear models do seem related to the processes being represented by the terms (e.g., U for uptake, R for regeneration, D for death), not all of the connections are clear (e.g., J for scavenging, L for uptake that gets exported). This makes is hard for the reader to keep track of which terms mean what. I suggest adding a table that describes, in terms of processes, what each symbol denotes.

Have you considered how a particular instance of the nonlinear model would respond to a substantial change to aeolian input, such as would happen in the LGM or a future climate change scenario. Does the technique presented shed light on how the nonlinear model would respond to this change in forcing? This could be mentioned in the Discussion section.

---

## Referee Comment (RC2) · J. Lauderdale (Referee) · 19 Sep 2018

J. Lauderdale (Referee)

jml1@mit.edu

In this manuscript, Pasquier and Holzer present a series of diagnostics to document the "life cycle" of dissolved iron in the ocean. Depending on the total source of iron (from an ensemble of nearly 300 solutions that "equally well" resemble oceanic nutrient distributions) they find that the majority of iron molecules are scavenged permanently from the ocean before they have had a chance to be biologically utilized. Of those that are taken up by phytoplankton, the majority will only have one circuit of the "ferrous wheel" before they too are permanently buried in the sediments.

I thought this was a really interesting paper, that certainly fits the criteria for publication

in Biogeosciences. I would like to suggest a few points that the authors might consider:

1.) Although fairly well written overall, in places I found the manuscript overly technical. For example, on page 9: lines 14-19 where there are 4 equivalences in as many lines, and only the last one (or two) are relevant. Perhaps there is a way to simplify? Furthermore, I appreciated where the authors had split their prose to identify the "physical" cause or effect and then the "mathematical" proof (page 7: line 20-22). Can this clarity be afforded elsewhere in the manuscript?

2.) Adding to the slightly overwhelming number of symbols used in this manuscript, I did come across something that looks like a mistake, or maybe requires clarification: Figure 1 suggests that "D" is the reversible scavenging process – after the first regeneration, iron is transported to the near-surface where it is scavenged onto a sinking particle and released at depth to be then transported into the euphotic layer and biologically utilized. Similarly, "D∼" is used for future reversible scavenging. However, section 2.3 defines "D" as "iron scavenging minus redissolution of scavenged iron" and the "permanent loss of iron due to burial in the sediments" (page 4: lines 6-7), which appears to correspond to "d" in the schematic.

3.) The phrase "Southern Ocean nutrient trapping" is frequently used, and I wondered if the authors could check that all uses are appropriate. For example, page 7: lines 17-19, I think the authors have the correct explanation that hydrothermal iron is added to density classes that upwell in the Southern Ocean, but is this really "nutrient trapping" and not just transport?

4.) Another paper that considered the iron fertilization efficiency was Dutkiewicz et al. (2006; GRL; doi: 10.1029/2005GL024987). Using an adjoint of the MITgcm biogeochemistry model, they found a similar pattern of tropical-Pacific-dominated primary production and carbon uptake when iron is added to the ocean.

5.) Finally, I wonder if the authors could comment on the caveat that their biogeochemical model may not capture the full array of interactions that might lead to en-

hanced iron regenerations through grazing by zooplankton, or bacteria/virus interactions, for instance. This is in regards to the "ferrous wheel" idea where recycling of iron is considered important (e.g. Kirchman (1996, Nature, doi: 10.1038/383303a0; Maldonado et al., 2005, GBC, doi: 10.1029/2005GB002481; Strzepek et al., 2005, GBC, doi: 10.1029/2005GB002490; Boyd et al., 2017, Nature Geoscience, doi:10.1038/ngeo2876). Maybe these views can be reconciled, with reference to figure 3?

---

## Referee Comment (RC3) · C. Völker (Referee) · 25 Sep 2018

Quite seldom, when reviewing manuscripts, I encounter a study where I wished I had had the idea myself. This manuscript, of which I have already reviewed an earlier version for another journal, is one of them. It introduces several new diagostics for the iron cycle that help to understand the fate of iron stemming from different external sources as it is advected through the ocean and every now and then gets taken up by a phytoplankton cell. Over the last decade or so it has been realized that other sources of iron besides dust (sediments, hydrothermal sources, volcanos,...) contribute substantially to the inventory of iron in the ocean, and estmates of iron's residence

time had to be corrected downwards. But there remain major uncertainties on the relative magnitude of these sources, and models with very different source strengths and residence times have been equally, albeit only moderately, successful in modelling the measured iron distribution. At this state of things it is an important step forward to have tools at hand that allow to quantify the biological effect of different iron sources, even if it is only for one specific iron model and assumes steady state.

The new diagnostics are based on a linearization of optimized steady-state solutions from a global ocean biogeochemical model to separate the effect of the different iron sources. The idea is to take the resulting iron distribution, and to linearize the nonlinear parts in the evolution equation for iron around that solution. The linearized equations are not only the basis for separating the effect of the different iron sources, but also allow, by splitting the linear operators, to separate in a second step, how often a specific concentration of iron has passed through biological uptake and remineralization, and how often it will do so in the future.

It is important that this exercise is not done for one specific model run, but for a whole family of model runs, differing mostly in the relative strength of iron sources, and correspondingly, in the timescale for scavenging, all reproducing iron observations about equally well (or badly). Of course this means that the linearized equations are different for every member of the ensemble, and the results obtained that way also differ to some extent. Nevertheless, the authors show quite convincingly that some results are quite robust and consistent between the different ensemble members.

The main results obtained in that way are that for reasonable strength of external iron sources, the average number of past and future passages through biological uptake of a given iron concentration is less than one, meaning that most iron has not or will not pass through biology before getting scavenged. This number is significantly higher for dust-deposited iron, since it enters the ocean closer to the place of biological uptake. Regardless of source the pattern of iron concentration that has passed through several uptake-recycling cycles converges towards a Southern Ocean-trapped pattern. Not all
results are equally important or surprising: for example the result that 'total DFe is more likely to have been regenerated in the past than it is to be regenerated in the future' (page 17, line 30) is fairly obvious from the fact that remineralisation happens deeper in the water column than e.g. dust deposition.

Some results may also have to do with the specific iron model: that iron fertilization is most efficient in the equatorial Pacific may also have to do with the particular parameterisation of a variable Fe:P quota in the model by Pasquier and Holzer, which basically follows a Michaelis-Menten-type curve, meaning that for Fe tending towards zero, the Fe:P ratio will also tend towards zero; the linearisation of the iron uptake would then ascribe a very high ratio of P to Fe export in regions with very low Fe. Most iron models produce an extremely low surface Fe concentration in the equatorial Pacific, which is far away from dust sources and where the upwelling waters are quite old, meaning thy are low in Fe.

The dependency on Fe:P is mentioned on page 22, line 11, and the sensitivity of the results on this is discussed briefly in the subsequent paragraph. In the discussion section this is however, discussed maybe a bit too briefly (page 25, line 30).

In summary I think that this is quite a significant paper for understanding the iron cycle in the ocean, and it should be published in Biogeosciences after minor revision.

The paper is quite well written (although it probably appeals more to a reader with some background in linear algebra) and I have checked the mathematical argumentation in depth and it is clear and correct. The authors attempt, and usually succeed in connecting the mathematically rigorous description of their results with what these results mean in terms of biogeochemistry. Nevertheless, here and there, the authors could do a bit more to make the explanations more palatable to the readership of Biogeosciences.

One example that I have is in the beginning of the section on future passages through the biological pump, where the authors explain that "the natural way to formulate the

necessary equations is to consider the time-reversed adjoint flow . . . The adjoints are defined for the volume-weighted inner product." While this is probably clear to a mathematically trained physicist, it may less be so for the average reader of Biogeosciences. Maybe the authors could add a few lines here on what the adjoints are, what the inner product.

My small remarks to the earlier version of this manuscript have been taken into account already, so I stop here.

---

## Author Response (AR1)

**Point-by-point Reply to all Referees**

November 1, 2018

Biogeosciences Discussion Manuscript `bg-2018-379` (submitted 14 Aug 2018):
**Iron fertilization efficiency and the number of past and future regenerations of iron in the ocean**

Authors:
**Benoît Pasquier** and **Mark Holzer**

Interactive discussion available at:
https://www.biogeosciences-discuss.net/bg-2018-379/

Dear Dr. Middelburg,

Please find the comments from the three referees and our point-by-point replies from the interactive discussion reproduced below. Along with our replies, we have added to this letter in green the corresponding revisions that were made to the manuscript.

The manuscript version with tracked changes, produced with the latexdiff package, is appended and shows the deleted parts struck through in red and the revised parts in blue.

In addition, we have re-arranged the title of our manuscript to read

> **The number of past and future regenerations of iron in the ocean and its intrinsic fertilization efficiency**

so that it better reflects the paper's content and emphasis.

Sincerely,

Benoît Pasquier and Mark Holzer

**Reply to Anonymous Referee #1**

**Referee #1 General Comments:** This is a review of "Iron fertilization efficiency and the number of past and future regenerations of iron in the ocean" by B. Pasquier and M. Holzer. The authors present a novel technique to track the life-cycle of dissolved iron (DFe) in the ocean. The authors apply the technique to multiple instances of data-constrained representations of the iron cycle and investigate how many cycles DFe parcels experience before and after participating in the biological pump. The authors use their technique to quantify the efficiency of iron fertilization on biological export, one of the motivations for their study. The manuscript is fairly well written, laying out clearly their novel technique, and they present well-designed experiments to utilize their technique. I think this manuscript will be a useful addition to the literature. I am making suggestions for some minor changes and/or additions.

**Authors' response:** We thank Referee #1 for these general comments.

*No changes to the manuscript in response to these general comments.*

No changes.

**Referee #1 Minor Point 1:** The abstract doesn't mention that the model used by the authors includes multiple types of external sources of DFe. I think it would be useful to mention in the abstract the types of external sources of DFe under consideration.

**Authors' response:** We agree that it would be good to mention the modelled external iron sources in the abstract.

*In response, we will revise the abstract to explicitly state that aeolian, sedimentary, and hydrothermal iron sources are modelled.*

The abstract now starts with:

> Iron fertilization is explored by tracking dissolved iron (DFe) through its life cycle from injection by aeolian, sedimentary, and hydrothermal sources (birth) to burial in the sediments (death).

**Referee #1 Minor Point 2:** The notation for the nonlinear model in Section 2.1 deviates from the notation in the author's previous work (Pasquier & Holzer, Biogeosciences, 2017) (e.g., removal of the Redfield ratio for the uptake terms, and changed notation for the particle transport terms). I suggest that the authors either use the same notation as their previous work, or describe how and why the notation in the current work differs from the previous work.

**Authors' response:** Referee #1 is correct that we changed notation from what we used to describe our fully coupled Fe–P–Si model [*Pasquier and Holzer*, 2017]. Here, we use simplified notation for extra clarity and readability, exploiting the fact that not all the complexity of the fully coupled model is needed to develop our new iron-cycle diagnostics.

*In response, we will add a brief statement to Section 2.1 which points this out and makes the connection with the corresponding symbols in* Pasquier and Holzer, *[2017].*

We have added the corresponding symbols of *Pasquier and Holzer* [2017] in the new Glossary (Appendix A, Table A1), which is referred to in an added parenthetical

sentence at the end of the paragraph describing Equation (1):

> (Note that we have simplified the notation of the fully coupled nonlinear model of Pasquier and Holzer (2017) for clarity and readability — see Appendix Table A1 for the symbol correspondence.)

**Referee #1 Minor Point 3:** There is no motivation given for the definition of the equivalent linear model in Section 2.2. I think the paper would benefit from having a paragraph describing what the goals/requirements of the equivalent linear model are, and how goals/requirements lead to the model that the authors are using.

**Authors' response:** The opening paragraph of Section 2.3 ("In order to track iron from its birth at the source to its eventual death. . .") was meant to motivate the need for the equivalent linear model, but we agree with Referee #1 that we do not explicitly point out why the nonlinear model itself cannot be used for tracing *partitions* of DFe or, equivalently, iron *labels*. The iron labels, unlike iron itself, are passive tracers and obey a linear equation of motion. Mathematically, only these linear tracers allow for correct partitioning of the DFe distribution because the superposition principle only applies to linear systems. This is often overlooked in the bigeoscience literature where studies typically evaluate the contribution of a given process by computing the anomaly that results from the removal of the process (e.g., removing a source). Such an anomaly approach results in errors that scale with the degree of nonlinearity of the system. For example, *Holzer et al.* [GBC, 2016] showed that omitting an iron source to evaluate its contribution to the DFe distribution underestimates the true contribution (evaluated using the equivalent linear model) by a factor of $\sim 2$. For the iron cycle, the nonlinearities of the uptake and scavenging processes are the reason why an equivalent linear model is required. *In response, we will add a paragraph to the beginning of Section 2.3 that includes the arguments above to explain the necessity of the equivalent nonlinear model.* We have revised the paragraph starting Section 2.3 to now read:

> In order to track iron from its birth at the source to its eventual death (via the irreversible part of scavenging), we consider a labelling tracer that we can think of as being attached to the nonlinearly evolving DFe. This labelling tracer has the same concentration as DFe but satisfies a linear evolution equation in which the nonlinear uptake and scavenging are replaced by linear processes. These linear processes are diagnosed from the nonlinear steady-state solution of Eq. (1) to provide identical uptake and scavenging rates and hence identical tracer solutions. It is necessary to employ such iron labelling tracers because their linear equations satisfy the superposition principle, which allows us to rigorously partition the iron concentration according to source type, number of regenerations, and so on. (The underlying parent model cannot directly be used for this purpose because of its nonlinearities.)

**Referee #1 Minor Point 4:** While some symbols chosen for the various terms in the nonlinear and equivalent linear models do seem related to the processes being represented by the terms (e.g., $U$ for uptake, $\mathcal{R}$ for regeneration, $\mathcal{D}$ for death), not all of the connections are clear (e.g., $J$ for scavenging, $\mathcal{L}$ for uptake

that gets exported). This makes is hard for the reader to keep track of which terms mean what. I suggest adding a table that describes, in terms of processes, what each symbol denotes.

**Authors' response:** We strongly agree with Referee #1 that good notation is important, and when possible we do use symbols whose meaning is self-evident. However, the desire for simple notation needs to be balanced with precision and clarity. For example, we need to use different symbols for linear and nonlinear operators, and we do not want to use an "s"-based symbol for both particle transport ("sinking") and scavenging. We think the notation in our manuscript is a reasonable compromise. We agree that a table describing our symbols could be helpful to the reader.

*In response, we will add a short glossary of symbols as suggested by Referee #1.*
We have added a glossary of symbols in new Appendix A.

**Referee #1 Minor Point 5:** Have you considered how a particular instance of the nonlinear model would respond to a substantial change to aeolian input, such as would happen in the LGM or a future climate change scenario. Does the technique presented shed light on how the nonlinear model would respond to this change in forcing? This could be mentioned in the Discussion section.

**Authors' response:** Thank you for the suggestion. We think that exploring the response of the iron cycle to changes in iron input is a rich subject deserving a separate study (and beyond the scope of the current manuscript). The diagnostics developed here probe the iron cycle without perturbing it and are therefore by themselves insufficient to infer the response to source changes. To examine the steady-state response to source changes, e.g., aeolian input for an LGM or future climate scenario, one would first have to use the fully coupled Fe—P—Si model of *Pasquier and Holzer* [2017] to solve for the perturbed steady-state nutrient cycles. Once these have been calculated, our diagnostics can readily be applied to the perturbed iron cycle to elucidate, for example, what the mean number of iron passages through the biological pump was during the LGM.

*In response, we will add some discussion to the Discussion and Caveats section to explain that while we diagnosed the unperturbed iron cycle, one can also apply the diagnostics to perturbed states resulting from added iron, e.g., to shed light on paleo or future climate scenarios.*
We have added the following sentence to the last paragraph of the Conclusions section:

> We also note that the approach of using a linear equivalent linear model to partition iron and diagnose its life cycle can also be applied to perturbed states (e.g., due to iron addition or changes in circulation) to shed light on how the iron cycle operates for various paleo or future climate scenarios. Finally, the concepts and methods employed here can be applied to other nutrients for a more complete picture of how the interaction between the biological pump and the physical transport shapes their distributions and cycling rates; we plan to do so in future work.

**Reply to Dr. Lauderdale (Referee #2)**

**Dr. Lauderdale's (Referee #2) General Comments:** In this manuscript, Pasquier and Holzer present a series of diagnostics to document the "life cycle" of dissolved iron in the ocean. Depending on the total source of iron (from an ensemble of nearly 300 solutions that "equally well" resemble oceanic nutrient distributions) they find that the majority of iron molecules are scavenged permanently from the ocean before they have had a chance to be biologically utilized. Of those that are taken up by phytoplankton, the majority will only have one circuit of the "ferrous wheel" before they too are permanently buried in the sediments. I thought this was a really interesting paper, that certainly fits the criteria for publication in Biogeosciences. I would like to suggest a few points that the authors might consider.

**Authors' response:** We thank Dr. Lauderdale (Referee #2) for these positive remarks.

*No changes to the manuscript in response to the General Comments.*
No changes.

**Referee #2 Minor Point 1:** Although fairly well written overall, in places I found the manuscript overly technical. For example, on page 9: lines 14–19 where there are 4 equivalences in as many lines, and only the last one (or two) are relevant. Perhaps there is a way to simplify? Furthermore, I appreciated where the authors had split their prose to identify the "physical" cause or effect and then the "mathematical" proof (page 7: line 20–22). Can this clarity be afforded elsewhere in the manuscript?

**Authors' response:** We think the equations in this instance are useful and should not be avoided because they define and clarify convenient notation (the $\chi$-weighted global averages). This notation is used later on multiple occasions in both the manuscript and figures and enhances subsequent readability. However, we do agree with Referee #2 that our manuscript need not be overly technical.

*In response, we will rephrase the second last sentence of this passage using more succinct expressions to ensure its point is clearly made with a minimum of symbols:* "Note that this fraction can be considered to be the $\chi_k$-weighted global average of the local unused fraction $f_k^{0\downarrow} \equiv \chi_k^{0\downarrow}/\chi_k$. This weighted average is defined as $\langle f_k^{0\downarrow} \rangle_{\chi_k} = \langle f_k^{0\downarrow}\chi_k \rangle / \langle \chi_k \rangle$, where we introduced the $\langle \cdot \rangle_{\chi_k}$ notation, which will be used throughout."
*We will check the revised manuscript everywhere for clarity and employ the "split-prose" approach where appropriate.*
We have revised the passage, which now reads:

> To quantify the amount of iron that was not regenerated in the past, we now ask how the unused fraction of the global DFe inventory varies with total iron source strength, $\sigma_{\text{tot}}$. Note that this fraction can be considered to be the $\chi_k$-weighted global average of the local unused fraction $f_k^{0\downarrow} \equiv \chi_k^{0\downarrow}/\chi_k$. This weighted average is defined as $\langle f_k^{0\downarrow} \rangle_{\chi_k} = \langle f_k^{0\downarrow}\chi_k \rangle / \langle \chi_k \rangle$, where we introduced the $\langle \cdot \rangle_{\chi_k}$ notation, which will be used throughout. The unused fractional DFe inventory regardless of source type is given by $\langle f^{0\downarrow} \rangle_{\chi}$, where $f^{0\downarrow} \equiv$

$\sum_k \chi_k^{0\downarrow}/\chi.$

**Referee #2 Minor Point 2:** Adding to the slightly overwhelming number of symbols used in this manuscript, I did come across something that looks like a mistake, or maybe requires clarification: Figure 1 suggests that $\mathcal{D}$ is the reversible scavenging process — after the first regeneration, iron is transported to the near-surface where it is scavenged onto a sinking particle and released at depth to be then transported into the euphotic layer and biologically utilized. Similarly, $\widetilde{\mathcal{D}}$ is used for future reversible scavenging. However, section 2.3 defines $\mathcal{D}$ as "iron scavenging minus redissolution of scavenged iron" and the "permanent loss of iron due to burial in the sediments" (page 4: lines 6–7), which appears to correspond to $d$ in the schematic.

**Authors' response:** We did try to minimize the number of symbols and always introduce new notation only to make the manuscript clearer or to provide precision where we think it is important. However, we agree with Referee #2 that we should have been clearer when introducing the operator $\mathcal{D}$. (The details of $\mathcal{D}$ are provided in Appendix A.) $\mathcal{D}$ is the linear integral operator that, applied to the DFe concentration field $\chi$, gives the local rate of scavenging minus the local rate with which scavenged iron is redissolved. The net local rate $(-\mathcal{D}\chi)(\boldsymbol{r})$ in Equation (2) can thus be locally positive (net DFe re-dissolution) but its vertical integral is always negative (a finite fraction is scavenged out of the system). Hence, $\mathcal{D}$ provides both the scavenging-pump transport for DFe (conservative "reversible scavenging") as well as the permanent DFe sink (non-conservative "death"). The field $d$ is the local iron death rate, which is calculated from $\mathcal{D}\chi$ as detailed in Appendix A and introduced later in Section 4.1. $d(\boldsymbol{r})$ is the rate at which DFe is removed at $\boldsymbol{r}$ by scavenging *and* instant sediment burial. Thus, $d$ (unlike $\mathcal{D}$) does not capture the recycling of scavenged DFe.

*In response, we will revise the text to clarify the action of $\mathcal{D}$ where it is introduced. For clarity and simplicity, we will also revise Figure 1 sightly by removing the adjoint operator symbols (they come too early for this introductory schematic) and to use the same type of $\mathcal{D}$-labelled arrow for both mid-stream reversible scavenging and for permanent burial, as $\mathcal{D}$ accomplishes both. We will also label the uptake process in the euphotic zone with $\mathcal{L}$, so that all physical processes are labelled on the figure.* We have reworded the description of $\mathcal{D}$ in the manuscript after Equation (2) to read:

..., and $\mathcal{D} \equiv \sum_j (1 - \mathcal{S}_j)\gamma_j$ is the reversible scavenging operator. More precisely, $\mathcal{D}$ is the linear integral operator that, applied to the DFe concentration field $\chi$, gives the local rate of scavenging minus the local rate with which scavenged iron is redissolved. Thus, $\mathcal{D}$ provides both the transport of the "scavenging pump" (conservative "reversible" scavenging) as well as the permanent iron sink due to burial in the sediments (non-conservative "death").

We have accordingly revised Figure 1 and its caption.

**Referee #2 Minor Point 3:** The phrase "Southern Ocean nutrient trapping"

is frequently used, and I wondered if the authors could check that all uses are appropriate. For example, page 7: lines 17–19, I think the authors have the correct explanation that hydrothermal iron is added to density classes that upwell in the Southern Ocean, but is this really "nutrient trapping" and not just transport?

**Authors' response:** We think that we use "Southern Ocean nutrient trapping" correctly here because we are considering hydrothermal DFe that has already been regenerated once ($n = 1$). We agree with Referee #2 that hydrothermal DFe is first "just transported" to the Southern Ocean surface where the density layers of the hydrothermal vents outcrop, but subsequently part of this hydrothermal DFe is utilized and trapped in the Southern Ocean (the trapping mechanism is described in the cited references). The plots of Figure 2 for hydrothermal DFe (column 3) show that hydrothermal DFe with $n \geq 1$ is found in the Southern Ocean with the characteristic pattern of Southern Ocean nutrient trapping.

*In response, we will double check the revised manuscript to ensure that all other occurrences of "Southern Ocean nutrient trapping" are appropriate.*
No changes.

**Referee #2 Minor Point 4:** Another paper that considered the iron fertilization efficiency was *Dutkiewicz et al.* (2006; GRL; doi:10.1029/2005GL024987). Using an adjoint of the MITgcm biogeochemistry model, they found a similar pattern of tropical-Pacific-dominated primary production and carbon uptake when iron is added to the ocean.

**Authors' response:** We thank Referee #2 for reminding us about this paper. We agree that we should have referenced it and that it will provide additional interesting context for our study. It is important, however, to appreciate that the experiments described by *Dutkiewicz et al.* (2006) are finite-amplitude perturbations, while our study quantifies the "natural" fertilization efficiency of the *unperturbed* iron cycle.
*In response, we will add references to* Dutkiewicz et al. *(2006) where relevant.*
We have added the following sentences at the beginning of the introduction where we cover previous work:

> Dutkiewicz et al. (2006) used an adjoint technique and a model of the coupled carbon, phosphorus, and iron cycles to quantify the sensitivity of global biological production and air-sea carbon fluxes to local perturbations in the aeolian iron source. They found that both quantities were most sensitive to iron addition in the central and eastern tropical Pacific.

In the second paragraph of Section 5.1, we have added the following sentences in Section 5.1 (Export supported per unit DFe injection at $r$):

> Furthermore, the local efficiency $\overline{m}^{\mathrm{P}}(r)$ is defined regardless of whether an iron source is actually present at $r$. The efficiency $\overline{m}^{\mathrm{P}}(r)$ quantifies the global phosphorus export rate *per unit DFe source rate* at $r$. At all points $r$, even where there is no actual source in the system, $\overline{m}^{\mathrm{P}}(r)$ can be considered to be the "sensitivity" of the linear equivalent system to the insertion of the arbitrary test source $s_k(r)$: Equation (14) shows that $\overline{m}^{\mathrm{P}}(r)$ is the proportionality between the export response $\Phi_k^{\mathrm{P}}(r)$ and the test source $s_k(r)$.

> In this sense, the fertilization efficiency $\overline{m}^{\mathrm{P}}$ is a close, but distinct, cousin of the sensitivity to small-amplitude perturbations considered by Dutkiewicz et al. (2006)

We have also added the following sentences in Section 5.1 where we discuss the geographic pattern of the natural fertilization efficiency:

> Interestingly, Dutkiewicz et al. (2006) found that the sensitivity of their coupled model to aeolian iron addition is also largest in the central tropical and eastern Pacific. This suggests that the intrinsic fertilization efficiency of the unperturbed state is shaped by similar processes as the sensitivity to small-amplitude perturbations.

Finally, we have added a comparison of the magnitude of our intrinsic fertilization efficiency to the fertilization efficiency estimates of Dutkiewicz et al. (2006) at the of Section 5.1:

> The sensitivity of global production to perturbations in the local aeolian source estimated by Dutkiewicz et al. (2006) has a spatial distribution with a range of about $20\text{--}180\,\mathrm{g\,C\,m^{-2}\,yr^{-1}}$ for a $0.02\,\mathrm{mmol\,Fe\,m^{-2}\,yr^{-1}}$ perturbation. The model of Dutkiewicz et al. (2006) exports 1/3 of its production as POP so that these sensitivities translate to a POP export per added DFe molecule of about $28\text{--}250\,\mathrm{mol\,C\,(mmol\,Fe)^{-1}}$, a lower bound on the full carbon export and about a third of the intrinsic fertilization efficiency estimated here. We emphasize that differences between these various estimates are not only due to uncertainties in the iron cycle (as expressed by our range of values), but also due to differences in the definition of fertilization efficiency, not to mention due to differences among models and methodologies.

**Referee #2 Minor Point 5:** Finally, I wonder if the authors could comment on the caveat that their biogeochemical model may not capture the full array of interactions that might lead to enhanced iron regenerations through grazing by zooplankton, or bacteria/virus interactions, for instance. This is in regards to the "ferrous wheel" idea where recycling of iron is considered important (e.g. *Kirchman*, 1996, Nature, doi:10.1038/383303a0; *Maldonado et al.*, 2005, GBC, doi:10.1029/2005GB002481; *Strzepek et al.*, 2005, GBC, doi:10.1029/2005GB002490; *Boyd et al.*, 2017, Nature Geoscience, doi:10.1038/ngeo2876). Maybe these views can be reconciled, with reference to figure 3?

**Authors' response:** We thank Referee #2 for bringing these studies to our attention. While the details of the ferrous wheel are beyond the scope of our study (as is their bearing on Figure 3), we agree that it would be appropriate to briefly reference these papers where we comment on the associated issues in relation to our model. Specifically, our simple formulation of the Fe:C uptake ratio may well be unrealistic (e.g., *Kirchman*, 1996; *Strzepek et al.*, 2005), as we acknowledged in the Discussion and Caveats section where we discuss the work of *Rafter et al.*, (2017). Similarly, we acknowledged (with reference to *Twining et al.*, 2014) that different remineralization lengthscales for iron and macronutrients (*Boyd et al.*, 2017) are not modelled. The effect of ligands on iron bioavailability (*Maldonado et al.*, 2005)

is also not represented by our model, which does not have dynamic ligands but instead prescribes a ligand distribution.

Importantly, we would like to note that not every process thought to operate on DFe needs to be explicitly modelled for a useful representation of the iron cycle. The inverse model of *Pasquier and Holzer* (2017) is of intermediate complexity, and any effect of the above issues is captured implicitly when parameters are optimized to fit the observed nutrient and phytoplankton fields. Explicit modelling of these complexities may be important for models that try to predict how the system will change in the future, but we do not think this is necessary for constraining and diagnosing the large-scale cycling of DFe in the current state of the ocean as we do here.

*In response, we will mention the effect of ligands on iron bioavailibility (which was missing in the submitted version), and we will add references to the suggested papers were we discuss the associated issues in the Discussion and Caveats section.*

We have added the following sentences on ligands in Section 6 (Discussion and caveats):

> Regarding ligands, the inverse model of Pasquier and Holzer (2017) prescribes an optimized distribution of a single type of ligand that is enhanced in hydrothermal plumes and old waters, with optimized parameters. Ligands are not dynamically transported and the effect of different ligand types on iron bioavailability (Maldonado et al., 2005) is neglected.

We have also revised the passage on the Fe:P stoichiometry in Section 6 to include the references suggested:

> A key control on our model results is the Fe:P stoichiometry. The model approximates the iron dependence of the Fe:P uptake ratio by a Monod function with a half saturation constant that is the same for all phytoplankton classes. We acknowledge that this may not be realistic. For example, Kirchman (1996) suggested that including the microbial "ferrous wheel", which operates with different stoichiometric ratios, could affect iron budgets in the euphotic zone. Similarly, Strzepek et al. (2005) explored variations in the iron regeneration rates of different organisms and cautioned modelers to pay careful attention to the details of the Fe:C stoichiometry. Relatedly, our model simply remineralizes iron in the same Fe:P ratio with which it was utilized, so that the vertical profiles of iron and phosphate remineralization have identical shapes. However, measurements by Twining et al. (2014) show that, at least for some phytoplankton species, iron is remineralized more slowly than phosphate, suggesting that our remineralization profile for iron could be too shallow. Different remineralization lengthscales for iron and phosphate were also emphasized by Boyd et al. (2017).

Finally, we have added a paragraph concluding the Discussion and caveats section to briefly discuss the role of complexity in models.

> Importantly, we would like to note that not every process thought to influence DFe needs to be explicitly modelled for a useful representation of the iron cycle. The model of Pasquier and Holzer (2017) is of intermediate

complexity, and effects due to processes not explicitly modelled are captured implicitly when parameters are optimized to fit the observed nutrient and phytoplankton fields. Explicit modelling of all known processes in their full complexity may be important for models that try to predict how the system will change in the future, but this is neither necessary nor desirable for constraining and diagnosing the large-scale cycling of DFe in the current state of the ocean as we do here.

**Reply to Dr. Völker (Referee #3)**

**Dr. Völker (Referee #3) Introductory Comment 1:** Quite seldom, when reviewing manuscripts, I encounter a study where I wished I had had the idea myself. This manuscript, of which I have already reviewed an earlier version for another journal, is one of them. It introduces several new diagostics for the iron cycle that help to understand the fate of iron stemming from different external sources as it is advected through the ocean and every now and then gets taken up by a phytoplankton cell. Over the last decade or so it has been realized that other sources of iron besides dust (sediments, hydrothermal sources, volcanos,...) contribute substantially to the inventory of iron in the ocean, and estmates of iron's residence time had to be corrected downwards. But there remain major uncertainties on the relative magnitude of these sources, and models with very different source strengths and residence times have been equally, albeit only moderately, successful in modelling the measured iron distribution. At this state of things it is an important step forward to have tools at hand that allow to quantify the biological effect of different iron sources, even if it is only for one specific iron model and assumes steady state.

**Authors' response:** We are delighted by Dr. Völker's (Referee #3) comments.

*No changes to the manuscript in response to these Introductory Comments.*

No changes.

**Referee #3 General Comment 1:** The new diagnostics are based on a linearization of optimized steady-state solutions from a global ocean biogeo-chemical model to separate the effect of the different iron sources. The idea is to take the resulting iron distribution, and to linearize the nonlinear parts in the evolution equation for iron around that solution. The linearized equations are not only the basis for separating the effect of the different iron sources, but also allow, by splitting the linear operators, to separate in a second step, how often a specific concentration of iron has passed through biological uptake and remineralization, and how often it will do so in the future.

**Authors' response:** Just to clarify, we do not linearize our nonlinear nutrient model in the usual sense, but instead construct an equivalent linear model for tracer labels as explained in our response to Referee #1, Minor Point 3. There is a fundamental difference: Linearization usually refers to the first-order Taylor expansion of the nonlinear model around a suitable base state, which captures the system behaviour for small perturbations about the base state. Here, we instead construct the equations for passive tracers that follow the DFe and that have the same solution as the nonlinear equations. The resulting model is linear in the sense that the labels participate in the physical processes in proportion to the local DFe abundance; hence all nonlinear processes are replaced by the DFe-label concentration multiplied by a rate coefficient diagnosed so that the linear and nonlinear models have identical solutions. The passive labelling tracers of the linear model then allow a rigorous partitioning according to source type, number of regenerations, and so on, because the superposition principle applies to the linear equivalent model.

*In response, we will add a sentence or two to the section on the linear equivalent model that cautions the reader not to mistake the linear model as a linearization of*

*the nonlinear model.*

We have added the following short paragraph at the end of Section 2.3 (Equivalent Linear Model: Iron Labelling Tracers):

> We caution the reader that the equivalent linear model (2) is not a linearization of the nonlinear parent model (1) in the usual sense. The equivalent linear model is constructed to partition DFe in the *unperturbed* system. By contrast, linearization usually refers to the first-order Taylor expansion of the nonlinear model around a base state, which captures the system behaviour for small perturbations about that base state.

**Referee #3 General Comment 2:** It is important that this exercise is not done for one specific model run, but for a whole family of model runs, differing mostly in the relative strength of iron sources, and correspondingly, in the timescale for scavenging, all reproducing iron observations about equally well (or badly). Of course this means that the linearized equations are different for every member of the ensemble, and the results obtained that way also differ to some extent. Nevertheless, the authors show quite convincingly that some results are quite robust and consistent between the different ensemble members.

**Authors' response:** Agreed.

*No change to the manuscript in response to this comment.*

No changes.

**Referee #3 Minor Point 1:** The main results obtained in that way are that for reasonable strength of external iron sources, the average number of past and future passages through biological uptake of a given iron concentration is less than one, meaning that most iron has not or will not pass through biology before getting scavenged. This number is significantly higher for dust-deposited iron, since it enters the ocean closer to the place of biological uptake. Regardless of source the pattern of iron concentration that has passed through several uptake-recycling cycles converges towards a Southern Ocean-trapped pattern. Not all results are equally important or surprising: for example the result that "total DFe is more likely to have been regenerated in the past than it is to be regenerated in the future" (page 17, line 30) is fairly obvious from the fact that remineralisation happens deeper in the water column than e.g. dust deposition.

**Authors' response:** We agree that not all our results are equally important and some may seem, after the fact, "fairly obvious". The likelihood of passing through the biological pump depends subtly on how transport to surface uptake from either external source or from internal regeneration samples the scavenging field. Our diagnostics allow this to be quantified rigorously for our model. The results can be rationalized after the fact, but we think that both qualitatively and certainly quantitatively, the finding of more likely past than future regeneration is far from obvious before one does the calculation. For example, it is not obvious that sedimentary iron, which has deep sources, is more likely to have passed through the pump in the past than in the future.

*No change to the manuscript in response to this minor comment.*

No changes.

**Referee #3 Minor Point 2:** Some results may also have to do with the specific iron model: that iron fertilization is most efficient in the equatorial Pacific may also have to do with the particular parameterisation of a variable Fe:P quota in the model by Pasquier and Holzer, which basically follows a Michaelis-Menten-type curve, meaning that for Fe tending towards zero, the Fe:P ratio will also tend towards zero; the linearisation of the iron uptake would then ascribe a very high ratio of P to Fe export in regions with very low Fe. Most iron models produce an extremely low surface Fe concentration in the equatorial Pacific, which is far away from dust sources and where the upwelling waters are quite old, meaning thy are low in Fe.
The dependency on Fe:P is mentioned on page 22, line 11, and the sensitivity of the results on this is discussed briefly in the subsequent paragraph. In the discussion section this is however, discussed maybe a bit too briefly (page 25, line 30).

**Authors' response:** We agree that the DFe dependence of the Fe:P uptake ratio is a major control on our model results. We would not characterize our discussion of the issue on page 22 as particularly brief: An entire paragraph (lines 13–21) is devoted to it. We agree, however, that we should reiterate the importance of the Fe:P ratio in the Discussion and Caveats section.

*In response, we will add a sentence or two to the Discussion and Caveats section reiterating the importance of the Fe:P ratio for shaping the pattern of the diagnosed natural iron fertilization efficiency.*

We have revised the passage in Section 6 (Discussion and caveats) to emphasize the importance of the Fe:P ratios:

A key control on our model results is the Fe:P stoichiometry. The model approximates the iron dependence of the Fe:P uptake ratio by a Monod function with a half saturation constant that is the same for all phytoplankton classes. We acknowledge that this may not be realistic. For example, Kirchman (1996) suggested that including the microbial "ferrous wheel", which operates with different stoichiometric ratios, could affect iron budgets in the euphotic zone. Similarly, Strzepek et al. (2005) explored variations in the iron regeneration rates of different organisms and cautioned modelers to pay careful attention to the details of the Fe:C stoichiometry. Relatedly, our model simply remineralizes iron in the same Fe:P ratio with which it was utilized, so that the vertical profiles of iron and phosphate remineralization have identical shapes. However, measurements by Twining et al. (2014) show that, at least for some phytoplankton species, iron is remineralized more slowly than phosphate, suggesting that our remineralization profile for iron could be too shallow. Different remineralization lengthscales for iron and phosphate were also emphasized by Boyd et al. (2017). Because our model is optimized to fit the DFe observations, with an emphasis on deep profiles relative to surface measurements Pasquier and Holzer (2017), a potentially too shallow remineralization of iron would be compensated by an increased strength of the biological pump. Furthermore, the relative amount of scavenging by opal and POP particles is optimizable in our model, so that deeper iron remineralization can be achieved by increasing

> the scavenging by opal. We acknowledge, however, that when optimizing the match to observed DFe, the model may produce biases in the relative contributions of the biological and scavenging pumps, which would affect our estimates of the number of passages through the biological pump.

**Referee #3 Minor Point 3:** In summary I think that this is quite a significant paper for understanding the iron cycle in the ocean, and it should be published in Biogeosciences after minor revision.

The paper is quite well written (although it probably appeals more to a reader with some background in linear algebra) and I have checked the mathematical argumentation in depth and it is clear and correct. The authors attempt, and usually succeed in connecting the mathematically rigorous description of their results with what these results mean in terms of biogeochemistry. Nevertheless, here and there, the authors could do a bit more to make the explanations more palatable to the readership of Biogeosciences.

One example that I have is in the beginning of the section on future passages through the biological pump, where the authors explain that "the natural way to formulate the necessary equations is to consider the time-reversed adjoint flow... The adjoints are defined for the volume-weighted inner product." While this is probably clear to a mathematically trained physicist, it may less be so for the average reader of Biogeosciences. Maybe the authors could add a few lines here on what the adjoints are, what the inner product.

My small remarks to the earlier version of this manuscript have been taken into account already, so I stop here.

**Authors' response:** We are delighted that Dr. Völker appreciates our work, and we certainly agree that our paper should be as accessible as possible to the Biogeosciences audience.

*In response, we will rephrase the introduction at the start of Section 4 (Future Contributions to Export) to explain and motivate why we use the adjoint, which is to track DFe backwards in time "from" death with computational efficiency.*

We have revised the passage on adjoints and the inner product at the start of Section 4 to read:

[revised manuscript text omitted]